# Octagons II:
# Strong Coupling

T. Bargheer$^{a,b}$, F. Coronado$^{c,d}$, P. Vieira$^{c,d}$

$^a$ *Institut für Theoretische Physik, Leibniz Universität Hannover,*
*Appelstraße 2, 30167 Hannover, Germany*

$^b$ *Deutsches Elektronen-Synchrotron DESY, Notkestr. 85, 22607 Hamburg, Germany*

$^c$ *Perimeter Institute for Theoretical Physics,*
*31 Caroline St N Waterloo, Ontario N2L 2Y5, Canada*

$^d$ *Instituto de Física Teórica, UNESP - Univ. Estadual Paulista,*
*ICTP South American Institute for Fundamental Research,*
*Rua Dr. Bento Teobaldo Ferraz 271, 01140-070, São Paulo, SP, Brasil*

till.bargheer@desy.de, fcidrogo@gmail.com, pedrogvieira@gmail.com

## Abstract

The octagon function is the fundamental building block yielding correlation functions of four large BPS operators in $\mathcal{N} = 4$ super Yang–Mills theory at any value of the 't Hooft coupling and at any genus order. Here we compute the octagon at strong coupling, and discuss various interesting limits and implications, both at the planar and non-planar level.

# 1 Introduction

The octagon function $\mathbb{O}(z, \bar{z}|\lambda)$ introduced in [1] provides a finite 't Hooft coupling representation for four-point correlation functions of large BPS operators, as recalled in Figure 1. The octagon is also the fundamental building block for these correlators beyond the planar limit [2]. In [3] the octagon was bootstrapped, providing an all-loop weak-coupling perturbative expansion, and in [4] a beautiful finite-coupling representation in terms of an infinite-dimensional Pfaffian was provided. In this small note, we study the octagon at strong coupling. Our study is split in two parts: The derivation of the strong-coupling result and its analysis.

The octagon is obtained by gluing two hexagon form factors together along one common edge. At weak coupling, each edge along which two hexagons are glued together becomes a "bridge" of planar propagator contractions between physical operators participating in the correlator. The number of propagators in a bridge is called the "bridge length", and it is a measure of the distance between the two adjacent hexagons. When the bridge length is asymptotically large, the two hexagons decouple. Beyond the asymptotic regime, one has to sum over a complete basis of virtual excitations (mirror magnons) that propagate

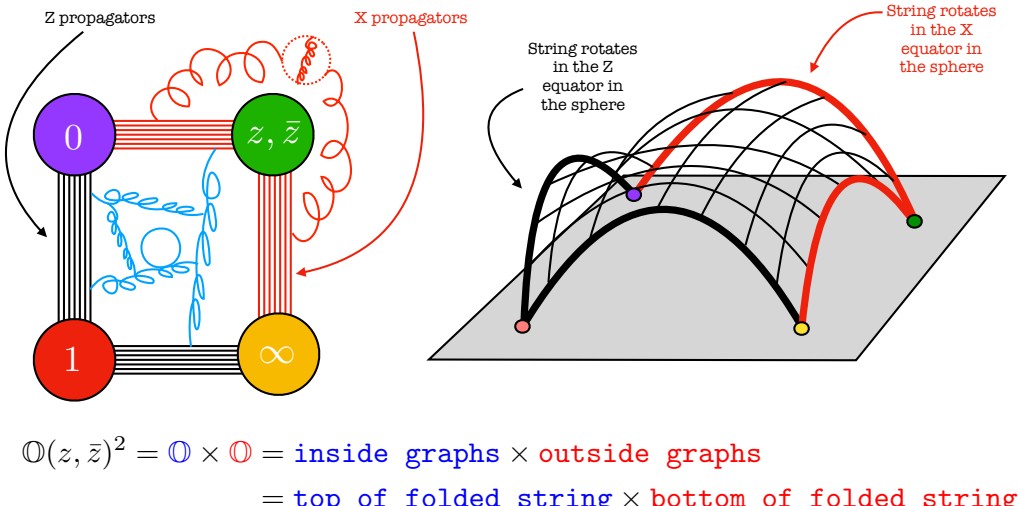

$$\mathbb{O}(z,\bar{z})^2 = \mathbb{O} \times \mathbb{O} = \texttt{inside graphs} \times \texttt{outside graphs}$$
$$= \texttt{top of folded string} \times \texttt{bottom of folded string}$$

**Figure 1:** We work here with the so-called *simplest* correlator introduced in [1]. Two operators are BPS primaries made out of only $Z$ or only $X$ fields respectively while the other two are BPS descendants composed of both $\bar{Z}$ and $\bar{X}$ so that at tree level there is a single square frame diagram describing this correlator. At loop level, for large operators, the inside and outside decouple. In string theory language the correlator is described by a folded string ending on some spinning geodesics [2]; the top and bottom folds decoupling is the string counterpart of the inside/outside gauge theory decoupling.

across the bridge between the two hexagons. These excitations capture the finite-size effects within the correlator, and computing their sum is difficult in general, especially when the mirror magnons can propagate across multiple bridges. In our case, the octagon is framed by asymptotically large bridges, and hence the mirror magnons are confined to a single bridge that splits the octagon into two hexagons. It turns out that this setup is very similar to the case of a three-point function between two BPS operators and one non-BPS operator, as illustrated in Figure 2. This case was considered by Komatsu, Kostov, Serban, and Jiang [5], and we can follow their "clustering" analysis almost verbatim.

In the three-point function case, the bridges between the non-BPS operator and the two BPS operators are taken to be large, such that the mirror magnons are confined to the single bridge that connects the two BPS operators to each other. The sum over mirror magnons is weighted by the transfer matrix of the non-BPS operator, which accounts for the interaction between the mirror magnons and the physical magnons on the non-BPS operator. If the third operator were also BPS, the correlator would be protected and the sum over mirror magnons would be trivial. The non-trivial weight breaks supersymmetry and thus leads to a non-trivial result. The authors of [5] have shown how to evaluate this "bottom-wrapping" non-trivial sum at strong coupling.

In the octagon case, the mirror magnons also live on a bridge connecting two BPS operators. This time, the opposite sides of the two hexagons do not connect to the same non-BPS operator, but to two *different* BPS operators. In order to perform the sum over mirror magnons, the two hexagons have to be brought to the same frame by a PSU$(2,2|4)$ transformation that maps the two different BPS operators onto each other. This change

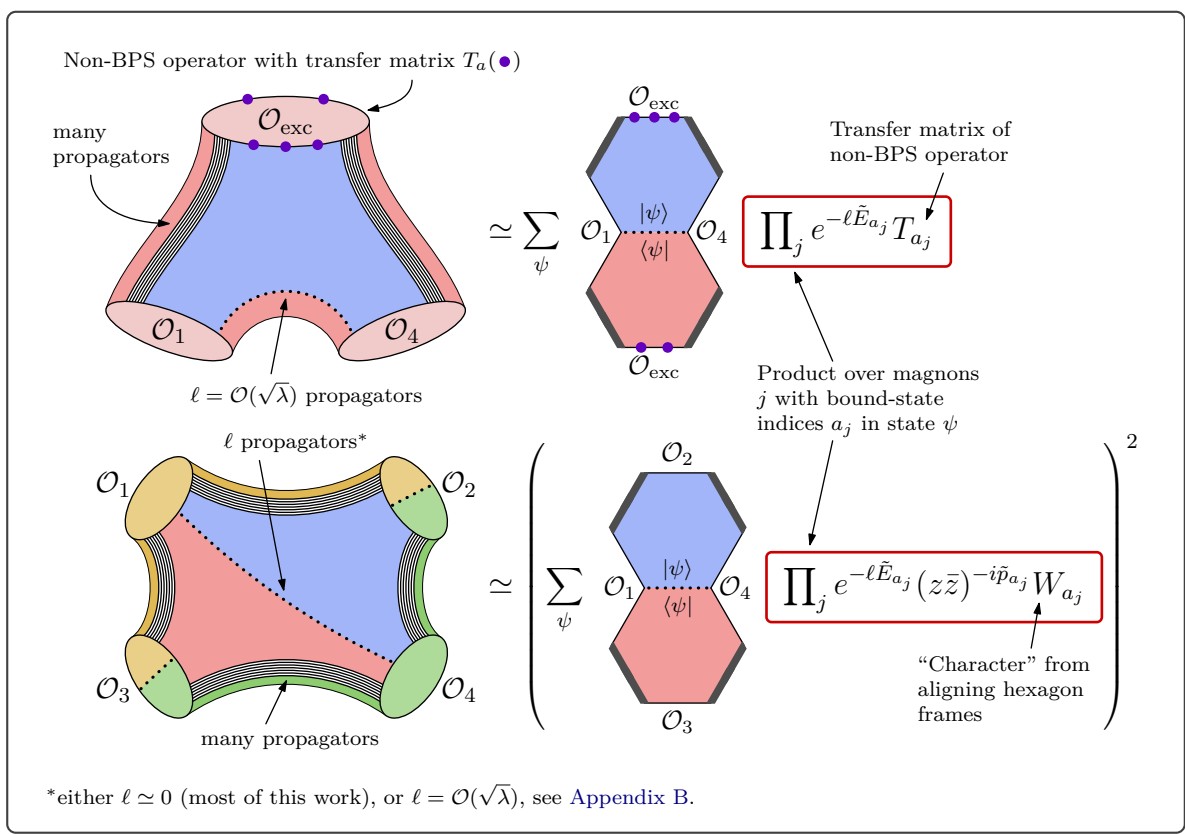

**Figure 2:** *Top:* Wrapping correction in the "bottom" channel of a three-point function between two BPS and one non-BPS operator; these were re-summed in [5]. *Bottom:* Virtual corrections for the "simplest" four-point function ($=$ octagon$^2$). We see that they exhibit strong similarities: Both sums run over the same states $\psi$, only the weights (red boxes) are different. The upper weight is a product of transfer matrix eigenvalues associated to the non-BPS operator; the lower weight is a product of characters associated to the cross-ratios of the four-point function. Under a simple replacement, we can thus re-use the clustering analysis of [5] to derive the strong-coupling octagon representation, see the final expression (2.21) below.

of frame induces a non-trivial character-like Boltzmann weight into the sum over mirror magnons. Again, this weight breaks supersymmetry and leads to a non-trivial result.

We thus see that the two cases are very similar at a technical level, and that is why we can recycle the analysis of [5] rather efficiently. We simply have to spot and replace the transfer matrices of the three-point case by the character weights of the octagon. An important part of the analysis in [5] was that the energy of mirror particles is constrained to be small, of order $1/\sqrt{\lambda}$, because they are multiplied by the length of their supporting bridge which is of order $\sqrt{\lambda}$. Here, $\lambda$ is the (large) 't Hooft coupling. In the octagon case, the mirror particles are weighted by their mirror momenta multiplied by (logarithms of) space-time cross ratios. The latter are naturally of order 1, and the mirror momenta is also of order 1 precisely when the mirror energy is of order $1/\sqrt{\lambda}$. Hence the kinematical regime in the octagon case is exactly the same as in the three-point function case.

Having realized this, the derivation exercise becomes rather straightforward and is presented in Section 2. The reader might want to skip directly to the final result, equation (2.21)

below. We observe a nice exponentiation as

$$\mathbb{O}(z, \bar{z}|\lambda) \simeq e^{-\sqrt{\lambda}\,\mathbb{A}(z,\bar{z})} \tag{1.1}$$

As explained in [2], $\mathbb{A}$ should be the minimal area of a string that ends on four BMN geodesics in AdS *and* rotates in the sphere, as sketched in Figure 1. The fact that the string moves in both AdS and the sphere makes it quite non-trivial to compute this minimal area from the string sigma model.[1] Still, the form of $\mathbb{A}$ clearly indicates that it should be possible to develop some Y-system like technology as in [6,7] to directly compute this minimal area at strong coupling, starting from the string sigma model. It would be very interesting to study this problem.

In Section 3, we analyze this result. We note that the area $\mathbb{A}$ is real and positive in the Euclidean regime where

$$z, \bar{z} = e^{-\varphi \pm i\phi} \tag{1.2}$$

with $\varphi$ and $\phi$ both real, and we explore what happens as we analytically continue the cross-ratios to various other interesting kinematical regimes, such as the Lorentzian regime and various OPE-like limits. We also make contact with [2], and explore the consequences of these results for the full non-planar expansion of the correlator of four large BPS operators. We conclude with some speculations and open problems in Section 4.

## 2 Derivation

As explained in the introduction, we can recycle the results of [5] to obtain the expression for the octagon function $\mathbb{O}$ at strong coupling. To see how this comes about, let us briefly review the construction of the octagon.

---

**Brief Review.** The octagon $\mathbb{O}_\ell$ is obtained by fusing two hexagon operators along a common mirror edge,

$$\tag{2.1}$$

The fusion amounts to summing and integrating over a complete basis of states $\psi$ on the common mirror edge. Such a basis is given by all $n$-magnon states, where each magnon is completely characterized by a rapidity $u$, a bound-state index $a$,[2] and indices $A, \dot{A}$ for the $a$'th antisymmetric representations of the two (left and right) $\mathfrak{su}(2|2)$ algebras [8,9]:

$$\mathbb{O}_\ell = \int [d\psi]\, \mu_\psi\, e^{-\tilde{E}_\psi \ell} \langle \mathcal{H}_1 | \psi \rangle \langle \psi | \mathcal{H}_2 \rangle\,, \qquad \int [d\psi] = \sum_{n=0}^{\infty} \frac{1}{n!} \sum_{\substack{a_i=1 \\ i=1..n}}^{\infty} \sum_{A_i,\,\dot{A}_i} \int du_1 \ldots du_n\,. \tag{2.2}$$

Each state in the sum is weighted by a measure factor $\mu_\psi$ for its creation, and a factor $e^{-\tilde{E}_\psi \ell}$ for its propagation across $\ell$ propagators, where $\tilde{E}_\psi$ is the (mirror) energy of $\psi$. In this work, we mostly consider $\ell = 0$, and hence $e^{-\tilde{E}_\psi \ell} = 1$.[3] Since we only study correlators of BPS operators, and since all outer mirror edges of the octagon will be occupied by a large number of propagators, there will be no (mirror) magnons on any of the outer edges of the two hexagons.

---

[1]In Appendix C, based on discussions with Martin Kruzcenski, we comment on how the simpler problem of minimal areas in AdS ending on geodesics – without any sphere – can often be solved.

The evaluation of a hexagon amplitude requires a choice of "frame", which is defined by the spacetime positions and R-symmetry orientations of the three BPS "vacuum" operators that attach to the three physical edges of the hexagon.[4] The two hexagons defining the octagon share two external operators (1 and 2 in (2.1)), but the third BPS operators attaching to the two hexagons are generically different (operators 3 and 4 in (2.1)). In order to consistently perform the sum over mirror states, the frames have to be aligned by a finite PSU(2, 2|4) transformation $g$ that maps one of these two different operators onto the other [9]: Choosing the first hexagon to be canonical $\mathcal{H}_1 = \hat{\mathcal{H}}$, the second hexagon is related to the first by conjugation with $g$: $\mathcal{H}_2 = g\hat{\mathcal{H}}g^{-1}$. The transformation $g$ is composed of a dilatation, a Lorentz rotation, and similar transformations in internal R-symmetry space. It is diagonal in the multi-magnon state basis, and hence amounts to a weight factor $\mathcal{W}_\psi$ in the sum over mirror states. The octagon function $\mathbb{O}_{\ell=0}$ therefore becomes

$$\mathbb{O} \equiv \mathbb{O}_{\ell=0} = \int [d\psi] \, \langle \hat{\mathcal{H}} | \psi \rangle \mu_\psi \mathcal{W}_\psi \langle \psi | \hat{\mathcal{H}} \rangle \,, \qquad \mathcal{W}_\psi = \langle \psi | g | \psi \rangle = e^{-i\tilde{p}_\psi \log(z\bar{z})} \, e^{iL_\psi \phi} \, e^{iR_\psi \theta} \, e^{iJ_\psi \varphi} \,. \quad (2.3)$$

Here, $\tilde{p}_\psi$ is the mirror momentum with $2i\tilde{p}_\psi = D_\psi - J_\psi$, and $D_\psi$, $L_\psi$, $R_\psi$ as well as $J_\psi$ are the charges of the state $\psi$ under the dilatation, Lorentz rotation, and the corresponding rotations in R-symmetry space that make up the transformation $g$, see [9]. For the case relevant to us, they take the values[5]

$$\theta = 0 \,, \qquad e^{-\varphi + i\phi} = z \,, \qquad e^{-\varphi - i\phi} = \bar{z} \,, \quad (2.4)$$

where $z$ and $\bar{z}$ as usual parametrize the spacetime cross ratios:

$$z\bar{z} = \frac{x_{12}^2 x_{34}^2}{x_{13}^2 x_{24}^2} \,, \qquad (1-z)(1-\bar{z}) = \frac{x_{14}^2 x_{23}^2}{x_{13}^2 x_{24}^2} \,. \quad (2.5)$$

The sum over magnon flavors $A_i$, $\dot{A}_i$ can be performed and gives [1]

$$\mathbb{O} = \int [d\psi]' \, (z\bar{z})^{-i\tilde{p}_\psi} W_{\{a_i\}} \mu_\psi \prod_{i<j} P_{a_i,a_j}(u_i, u_j) \,, \qquad \int [d\psi]' = \sum_{n=0}^{\infty} \frac{1}{n!} \sum_{\substack{a_i=1 \\ i=1..n}}^{\infty} \int du_1 \dots du_n \,, \quad (2.6)$$

where $P_{ab}(u, v)$ is a function of the rapidities $u$, $v$, and bound-state indices $a$, $b$, see (A.8) in [1]. The weight factor $(z\bar{z})^{-i\tilde{p}_\psi} W_{\{a_i\}}$ contains the "character" $W_{\{a_i\}}$. With (2.4), it takes the form

$$W_{\{a_i\}} = \prod_{j=1}^{n} W_{a_j} \,, \qquad W_a = -4 \sinh\left(\frac{\varphi}{2} + \frac{i\phi}{2}\right) \sinh\left(\frac{\varphi}{2} - \frac{i\phi}{2}\right) \frac{\sin(a\phi)}{\sin\phi} \,. \quad (2.7)$$

**Identification.** The formula (2.6) for the complete octagon sum closely resembles the "bottom wrapping" expression of [5] (see eqs. (5.2) and (5.3) there) that sums all mirror excitations on a mirror edge between two BPS operators in a three-point function with another non-BPS operator, see Figure 2. In that latter case, each state $\psi$ in the mirror state sum is weighted by the product $\prod_j T_{a_j}(u_j)$ over constituent magnons $j$ of $\psi$, with $a_j$ and $u_j$ being the $j$'th magnon's bound-state index and rapidity, and $T_a$ being the transfer matrix eigenvalue of the third (non-BPS) operator. Instead of this product over transfer matrix eigenvalues, our sum (2.6) is weighted by the factor $(z\bar{z})^{-i\tilde{p}_\psi} W_{\{a_i\}}$ that originates in the misalignment of the two "opposite" BPS operators of our four-point function. Importantly, this weight also factorizes into a product over the $n$ constituent magnons of $\psi$:

$$(z\bar{z})^{-i\tilde{p}_\psi} W_{\{a_i\}} = \prod_{j=1}^{n} (z\bar{z})^{-i\tilde{p}_{a_j}(u_j)} W_{a_j} \,. \quad (2.8)$$

This factorization is due to the additivity of the mirror momentum $\tilde{p}$ and the factorization of the character (2.7). We therefore recover the expression (5.3) of [5] for the bottom wrapping sum by identifying the transfer matrix eigenvalue $T_a(u)$ in that expression as $T_a(u) = (z\bar{z})^{-i\tilde{p}_a(u)} W_a$.[6] This straightforward identification at the level of individual magnons makes the "clustering" analysis of [5] also applicable to our case, and lets us directly re-use their final re-summed result (eqs. (5.49)

and (5.43) there)! To state the final result, we only need the strong-coupling expression of the mirror bound-state momentum

$$\tilde{p}_a(u) = u - g\left(\frac{1}{x^{[+a]}} + \frac{1}{x^{[-a]}}\right), \tag{2.9}$$

with the Zhukowsky variable $x(u)$ and the shorthand $x^{[\pm a]}$ being defined via

$$\frac{u}{g} = x + \frac{1}{x} \qquad \Rightarrow \qquad x(u) = \frac{u + \sqrt{u - 2g}\sqrt{u + 2g}}{2g}, \qquad x^{[\pm a]}(u) \equiv x(u \pm a\, i/2). \tag{2.10}$$

At strong coupling, (2.9) becomes

$$\tilde{p}_a(u) \simeq \frac{a\, u}{2\sqrt{4g^2 - u^2}} \qquad \text{for} \qquad g \to \infty. \tag{2.11}$$

With $\log(z\bar{z}) = -2\varphi$, we thus find at strong coupling

$$T_a(u) = (z\bar{z})^{-i\tilde{p}_a(u)} W_a \simeq \exp\left(a\varphi \frac{iu}{\sqrt{4g^2 - u^2}}\right) W_a. \tag{2.12}$$

In summary, we can immediately recycle the results of [5] with appropriate replacements of the non-BPS transfer matrices there by the characters produced by the two differently aligned hexagons in the four-point function. As explained above, a simple comparison of the starting point in [5] with the octagon infinite sum representation indicates that we should take

$$T_a = \overbrace{e^{a\varphi \frac{iu}{\sqrt{4g^2 - u^2}}}}^{(z\bar{z})^{\texttt{mirror momentum}}} \underbrace{X \sin(a\phi)}_{\texttt{character}}, \qquad X \equiv -\frac{\overbrace{4\sinh\left(\frac{\varphi}{2} + \frac{i\phi}{2}\right)\sinh\left(\frac{\varphi}{2} - \frac{i\phi}{2}\right)}^{a\texttt{-independent part of character } X = -2i(1-z)(1-\bar{z})/(z-\bar{z})}}{\sin(\phi)} \tag{2.13}$$

and plug it into the final strong-coupling expression (5.49) and (5.43) in [5]. This gives

$$\log \mathbb{O} \simeq \int_{-2g}^{+2g} \frac{du}{2\pi} \sum_{n=1}^{\infty} \frac{1}{n} \sum_{\{n_a\}} (-1)^{(K-1)} (K-1)! \prod_a \frac{(T_a)^{n_a}}{n_a!}, \tag{2.14}$$

where the sum over the positive mode numbers $n_a$ is constrained as $\sum_a a\, n_a = n$, and where $K$ is defined as $K \equiv \sum_a n_a$. What follows is a straightforward simplification of this expression. The reader might want to jump to the final simplified result (2.21).

---

[2] A magnon with bound-state index $a$ is composed of $a$ fundamental magnons.

[3] Our strong-coupling result remains correct for $l \neq 0$, as long as $\ell \ll \sqrt{\lambda}$. For $\ell \sim \sqrt{\lambda}$, the result gets modified, but is still calculable, see Appendix B.

[4] Three half-BPS vacuum operators are preserved by a common (diagonal) $\mathfrak{su}(2|2)$ algebra [10], which fixes the hexagon amplitude to a large extent [8].

[5] The octagon is most clearly isolated in correlators of the type considered in [1, 2], where two of the operators are BPS primaries $\mathcal{O}_1 = \mathrm{tr}(X^{2k})$, $\mathcal{O}_4 = \mathrm{tr}(Z^{2k})$, and two operators are BPS descendants $\mathcal{O}_2 = \mathcal{O}_3 = \mathrm{tr}(\bar{Z}^k \bar{X}^k) + \text{(permutations)}$. See Appendix A for more details and how this implies $\alpha = \bar{\alpha} = 1$.

[6] The bottom wrapping expression (5.3) in [5] is written as a sum over occupation numbers $n_a$ of bound-state indices $a$. Since all factors in the integrand of (2.6) and in [5] are symmetric under permutations of magnons with identical bound-state indices, it is easy to convert back and forth between a sum over the total number $n$ of constituent magnons and sums over occupation numbers $n_a$. In particular, the weight $(z\bar{z})^{-i\tilde{p}_\psi} W_{\{a_i\}}$ can equally be written as $(z\bar{z})^{-i\tilde{p}_\psi} W_{a_i} = \prod_{a=1}^{\infty} \prod_{j_a=1}^{n_a} (z\bar{z})^{-i\tilde{p}_a(u_{j_a}^a)} W_a$, where $u_1^a, \ldots, u_{n_a}^a$ are the rapidities of the $n_a$ magnons with bound-state index $a$, and $\tilde{p}_a(u)$ is the mirror momentum of the $a$'th bound state.

**Simplification.** The goal is to factorize the integrand in a way that we can decouple the sum over mode numbers $n_a$ into independent sums which we can then perform. For instance, to factorize the factorial factor we simply write $(K-1)! = \int dt\, e^{-t} t^{K-1}$ so that $K = \sum_a n_a$ appears in exponents and thus breaks apart into a product of factors for the various mode numbers. Next we want to cancel the $1/n$ factor in (2.14) which is easy by a simple integration by parts. For that we note that the rapidity $u$ dependence only arises through the exponential factor in $T_a$; using the definition of $n$ we can take it out of the $n$ sum completely and write

$$\frac{1}{n}\int_{-2g}^{2g} du\, e^{in\varphi u/\sqrt{4g^2-u^2}} = \frac{2g}{n}\int_{-\infty}^{\infty} d\theta\, \frac{d\tanh(\theta)}{d\theta} e^{in\varphi \sinh(\theta)} \qquad \Big| \; u = 2g\tanh(\theta)$$

$$= -2gi\varphi \int_{-\infty}^{\infty} d\theta\, \sinh(\theta) e^{in\varphi \sinh(\theta)} \qquad \Big| \; \text{integration by parts} \qquad (2.15)$$

$$= 2g\varphi \int_{-\infty}^{\infty} d\theta\, \cosh(\theta) e^{-n\varphi \cosh(\theta)} \qquad \Big| \; \theta \to \theta + i\pi/2 \; \text{shift} \qquad (2.16)$$

Here we assumed $\mathrm{Re}(\varphi) > 0$, otherwise we should pick an opposite shift in (2.16). We obtain the desired factorization

$$\log \mathbb{O} \simeq -\frac{g}{\pi}\,\varphi \int_{-\infty}^{\infty} d\theta\, \cosh(\theta) \int_0^{\infty} \frac{dt}{t} e^{-t} \sum_{n=1}^{\infty} \sum_{\{n_a\}} \prod_a \frac{\left(-Xt\sin(a\phi)\, e^{-a\varphi\cosh\theta}\right)^{n_a}}{n_a!}\,, \qquad (2.17)$$

where $n = \sum a\, n_a$. Adding and subtracting a $n = 0$ term (corresponding to all $n_a = 0$) we get a final factorization into *unconstrained* mode numbers as

$$\log \mathbb{O} \simeq -\frac{g}{\pi}\,\varphi \int_{-\infty}^{\infty} d\theta\, \cosh(\theta) \int_0^{\infty} \frac{dt}{t} e^{-t} \left[\prod_{a=1}^{\infty} \sum_{n_a=0}^{\infty} \frac{\left(-Xt\sin(a\phi)\, e^{-a\varphi\cosh\theta}\right)^{n_a}}{n_a!} - 1\right], \qquad (2.18)$$

We can now perform the sum over the mode numbers $n_a$,

$$\log \mathbb{O} \simeq -\frac{g}{\pi}\,\varphi \int_{-\infty}^{\infty} d\theta\, \cosh(\theta) \int_0^{\infty} \frac{dt}{t} e^{-t} \left[\exp\left(-\sum_{a=1}^{\infty} tX\sin(a\phi) e^{-a\varphi\cosh(\theta)}\right) - 1\right], \qquad (2.19)$$

The sum over $a$ can also be done, leading to

$$\log \mathbb{O} \simeq -\frac{g}{\pi}\,\varphi \int_{-\infty}^{\infty} d\theta\, \cosh(\theta) \int_0^{\infty} \frac{dt}{t} e^{-t} \left[e^{-tY(\theta)} - 1\right], \qquad (2.20)$$

where $Y(\theta)$ is given by (2.22) below. Finally, the integral over $t$ yields $-\log(1+Y)$ and so we obtain (2.21) once we recall the relation to the 't Hooft coupling $g = \sqrt{\lambda}/4\pi$.

**Result.** We thus find

$$\log \mathbb{O} \simeq \frac{\sqrt{\lambda}}{2\pi}\int_{-\infty}^{\infty} \frac{d\theta}{2\pi}\, \varphi \cosh(\theta) \log(1 + Y(\theta))\,, \qquad (2.21)$$

where

$$Y(\theta) = -\frac{\sin\left(\frac{\phi}{2} + i\frac{\varphi}{2}\right)\sin\left(\frac{\phi}{2} - i\frac{\varphi}{2}\right)}{\sin\left(\frac{\phi}{2} + i\frac{\varphi}{2}\cosh(\theta)\right)\sin\left(\frac{\phi}{2} - i\frac{\varphi}{2}\cosh(\theta)\right)}\,. \qquad (2.22)$$

We derived this result from the octagon expression as two fused hexagons. It would be very nice to derive it from the string world-sheet *a la* [6]. The TBA-like result (2.21) is very

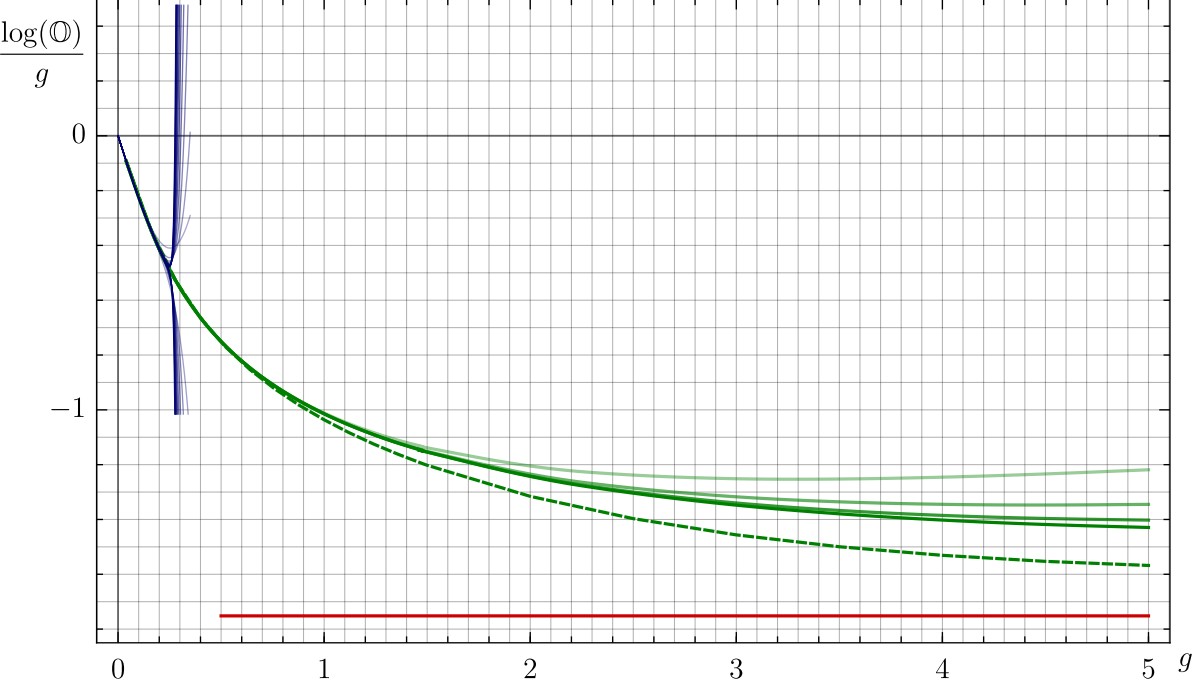

**Figure 3:** Comparison of various data for the ratio $\log(\mathbb{O})/g$ as a function of the coupling $g$. The thin blue lines that diverge near the radius of convergence at $g = 1/4$ are the perturbative results of [1], from two loops all the way to 20 loops. The solid green lines are the numerical evaluation of the determinant representation of [4]. The latter nicely agrees with the perturbative representation, and continues it beyond its radius of convergence. To evaluate the determinant, we truncated the semi-infinite matrix to sizes $N = 10, 15, 20, 25$ with darker lines corresponding to larger sizes; clearly it becomes more and more important to consider larger matrices at strong coupling. The dashed green line represents an extrapolation of these results towards infinite matrices (using the results from $N = 2$ to $N = 25$ and a simple fit $a + b/N$). Finally, the red horizontal line is the strong-coupling prediction in (2.21). In this plot, we use Euclidean cross-ratios $\varphi = 1/10$, $\phi = \pi/3$.

reminiscent of the type of expressions coming out in those papers from exploring the classical string integrability. Note in particular that

$$1 + Y = \frac{\sinh\left(\frac{\varphi}{2} - \frac{\varphi}{2}\cosh(\theta)\right)\sinh\left(\frac{\varphi}{2} + \frac{\varphi}{2}\cosh(\theta)\right)}{\sinh\left(\frac{i\phi}{2} - \frac{\varphi}{2}\cosh(\theta)\right)\sinh\left(\frac{i\phi}{2} + \frac{\varphi}{2}\cosh(\theta)\right)} \tag{2.23}$$

takes a very nice factorized form, which allows us to split the putative area in (2.21); perhaps one contribution will come from the AdS part and another from the sphere. We can also include a finite internal bridge length in the octagon (see Appendix B), and try to reproduce that more general result from the string world-sheet.

In Figure 3, we compare the strong-coupling result (2.21) to the finite-coupling representation of the octagon recently worked out in [4]. It looks consistent, but it would be very nice to work out the one-loop prefactor in (1.1), and to improve the determinant evaluation at strong coupling to perform a more conclusive comparison. Furthermore, if we could compute the one-loop prefactor from the octagon representation, it would provide us with yet another powerful data point to reproduce from the string sigma model.

The result (2.21) was derived for real $\phi \in [0, 2\pi]$ and real $\varphi > 0$ (see the shift (2.16)). This translates into $\bar{z} = z^*$ and $|z|, |\bar{z}| < 1$. Of course, we can (and will) move away and study any range of parameters – both real and complex – but we need to carefully analytically continue the result starting from this safe starting point. This is particularly obvious even if we remain in the fully Euclidean region where $\phi$ and $\varphi$ are both real. The $Y$-function in (2.22) is *even* under $\varphi \to -\varphi$, but because of the $\varphi$ outside the log in the area (2.21), the integrand is odd, and thus it seems that the full result is *odd*. That is wrong. The full area is *even*, nicely realizing the $z \leftrightarrow 1/\bar{z}$ symmetry of the octagon. To see that, however, is a bit non-trivial. It turns out that as we rotate from $\varphi > 0$ to $\varphi < 0$, infinitely many singularities hit the integration contour, which therefore needs to be re-arranged non-trivially. This produces an additional minus sign required to convert the *naive odd* guess into the *correct even* result. This contour re-arrangement illustrates very nicely the kind of manipulations involved in more general analytic continuations, and is presented in detail in Appendix D.1.

# 3   Analysis

In the Euclidean regime, the two cross-ratio variables $z$ and $\bar{z}$ are complex conjugate to each other, and therefore $\varphi$ and $\phi$ in (1.2) are real. Then the area

$$\log \mathbb{O} \simeq \frac{\sqrt{\lambda}}{2\pi} \int_{-\infty}^{\infty} \frac{d\theta}{2\pi} \, \varphi \cosh(\theta) \log \left[ 1 - \frac{\sin\left(\frac{\phi}{2} + i\frac{\varphi}{2}\right) \sin\left(\frac{\phi}{2} - i\frac{\varphi}{2}\right)}{\sin\left(\frac{\phi}{2} + i\frac{\varphi}{2}\cosh(\theta)\right) \sin\left(\frac{\phi}{2} - i\frac{\varphi}{2}\cosh(\theta)\right)} \right], \quad (3.1)$$

is manifestly real. The logarithm is negative and $\varphi$ is positive, thus the full integrated right hand side is negative. It is also manifestly periodic in $\phi$, leading to a single-valued expression in the Euclidean regime, as expected (see Figure 4). Since it is multiplied by a large string tension $\sqrt{\lambda}$, we see that the octagon is exponentially small in the Euclidean regime, and thus

$$\mathbb{O} \to 0 \quad \text{when} \quad \lambda \to \infty. \tag{3.2}$$

This has nice implications for the double-scaling limit of the "simplest" correlator of [1],[7] where the charges of the external operators scale as $\sqrt{N_c}$. This limit was analyzed in [2]. Since all configurations that involve the octagon $\mathbb{O}$ vanish due to (3.2), the full non-planar correlator in the Euclidean regime reduces to a sum of BMN-like configurations, in which the string world-sheet degenerates into various point-like geodesics. Such configurations first show up at genus one. All in all, these BMN configurations re-sum into the explicit expression (4.3) in [2].

The octagon expression (3.1) also exhibits a rich behavior in various interesting kinematical regimes, as we will now explore. To study these limits, it is useful to derive two mathematical formulae for our expression. The first is

$$\log \mathbb{O} \simeq -\frac{\sqrt{\lambda}}{4\pi^{3/2}} \sqrt{-\log(z\bar{z})} \Big( \mathrm{Li}_{3/2}(1) + \mathrm{Li}_{3/2}(z\bar{z}) - \mathrm{Li}_{3/2}(z) - \mathrm{Li}_{3/2}(\bar{z}) \Big), \tag{3.3}$$

it is valid in the Euclidean sheet with $|z\bar{z}| \le 1$, and for kinematics such that the small rapidity $\theta \ll 1$ region dominates the integral. The second expression reads

$$\log \mathbb{O} \simeq -\frac{\sqrt{\lambda}}{4\pi^2} \frac{\log(z/\bar{z})}{2i} \left( 2\pi - \frac{\log(z/\bar{z})}{2i} \right), \tag{3.4}$$

---

[7]See Section A for a brief review.

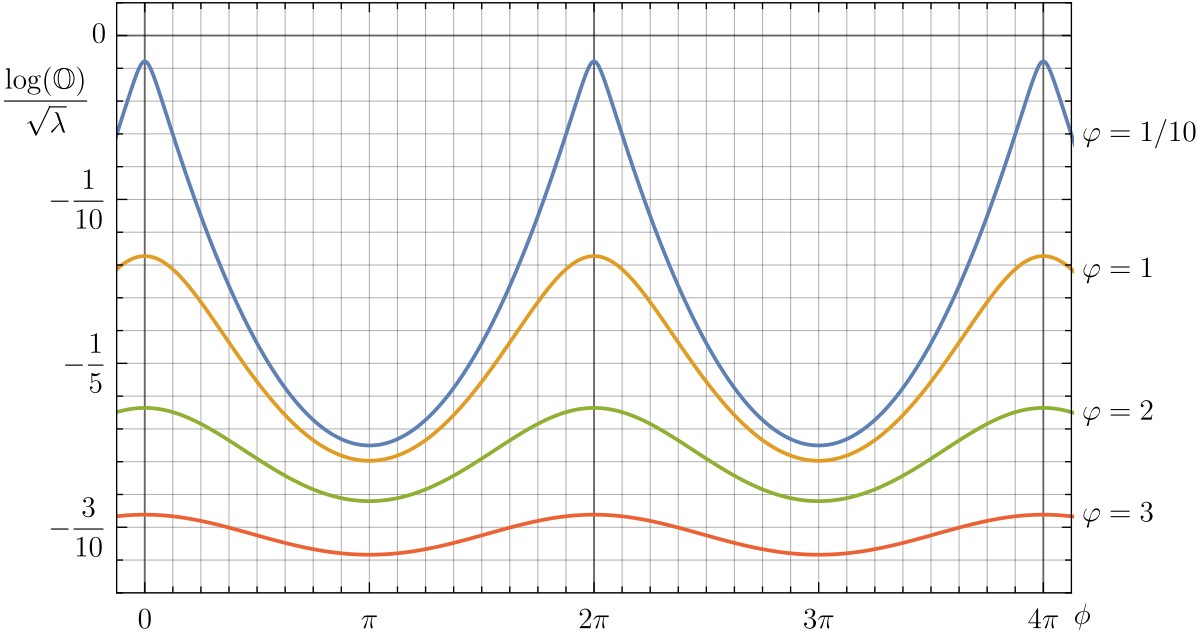

**Figure 4:** In the Euclidean regime, the area is single-valued on the physical sheet, i. e. it is periodic in the angle $\phi$. As a function of this angle, we see that it develops kinks as $\varphi \to 0$, and approaches a constant as $\varphi \to \infty$.

and it holds whenever the kinematics are such that it is the large $\theta \gg 1$ region which controls the integral. It turns out that in all OPE limits discussed below, it is indeed either the small or the large rapidity regions which control the octagon behavior, so these expressions will be all we need in the next section. In the following, we sketch the derivations of these two expressions:

**(3.3) derivation**: The kinematical limit in (3.3) is dominated by small $\theta$. In order to analyze limits of $z$ or $\bar{z}$ individually, we write the $Y$-function as:

$$Y = -\frac{\sinh\left(\frac{1}{2}\log z\right)\,\sinh\left(\frac{1}{2}\log \bar{z}\right)}{\sinh\left(\frac{1}{2}\log z + \frac{1}{2}\sinh^2\left(\frac{\theta}{2}\right)\log(z\bar{z})\right)\,\sinh\left(\frac{1}{2}\log \bar{z} + \frac{1}{2}\sinh^2\left(\frac{\theta}{2}\right)\log(z\bar{z})\right)}\,. \tag{3.5}$$

We see that when $\theta$ is real and $z$, $\bar{z}$ are either real or complex conjugate, then $-1 \leq Y \leq 0$. Moreover, we recognize that when a cross ratio $z$ or $\bar{z}$ approaches 0 or 1, then the $Y$–function (3.5) approaches $-1$ only in the region of very small rapidity $\theta$. Therefore, it is this region which dominates the integral in (2.21), so that the integrand

$$\cosh(\theta)\log\left(1 + Y\right) = \cosh(\theta)\log\frac{\left(1 - e^{\log(z\bar{z})\sinh^2(\theta/2)}\right)\left(1 - z\bar{z}\,e^{\log(z\bar{z})\sinh^2(\theta/2)}\right)}{\left(1 - z\,e^{\log(z\bar{z})\sinh^2(\theta/2)}\right)\left(1 - \bar{z}\,e^{\log(z\bar{z})\sinh^2(\theta/2)}\right)} \tag{3.6}$$

can be dramatically simplified by expanding $\sinh(\theta/2) \simeq \theta/2$ and $\cosh(\theta/2) \simeq 1$, and using

$$\int\limits_{-\infty}^{\infty} d\theta\,\log\left(1 - x\exp(-y\,\theta^2)\right) = -\sqrt{\pi/y}\,\mathrm{Li}_{3/2}(x) \tag{3.7}$$

to establish (3.3). From a thermodynamic Bethe ansatz context, these limits resemble non-relativistic limits, where particles have small rapidities.

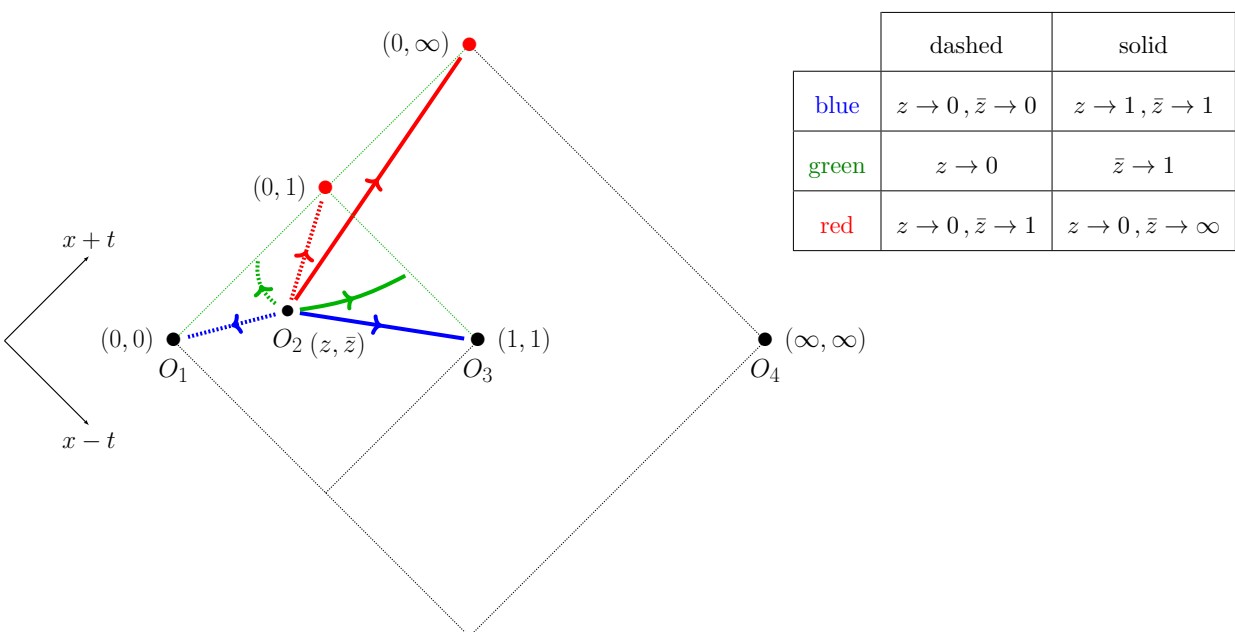

**Figure 5:** Poincaré patch in light-cone coordinates with operators $O_2$ at $(z, \bar{z})$ and $O_1, O_3, O_4$ in canonical positions. OPE limits are obtained when $O_2$ approaches any of the corners or edges of the dashed squares. We indicate in blue the Euclidean or space-like OPE's, in green the single light-like OPE's and in red the double light-like OPE's.

and

---

**(3.4) derivation**: The kinematical limit in (3.4) is dominated by large $\theta$. It is the relevant limit, for example, when $\log(z\bar{z}) \to 0$ so we see that all exponents in (3.6) can be effectively set to zero *unless $\theta$* is huge, of order $\log\log(z\bar{z})$. Furthermore, the full expression is multiplied by $\varphi = -\log(z\bar{z})/2$ and thus vanishes unless $\theta$ is huge indeed. So we can freely replace all hyperbolic functions by large $\theta$ single exponential terms. Once that is done, it suffices to use

$$\int\limits_{-\infty}^{\infty} d\theta\, e^{\theta} \log\left(1 - x\exp(-y\,e^{\theta})\right) = -\frac{1}{y}\,\mathrm{Li}_2(x) \tag{3.8}$$

to evaluate all resulting integrals and thus get that the area is proportional to $\mathrm{Li}_2(1) + \mathrm{Li}_2(z\bar{z}) - \mathrm{Li}_2(z) - \mathrm{Li}_2(\bar{z})$. Expanding for $\log(z\bar{z}) \to 0$ (i.e. for $z \to 1/\bar{z}$) does simplify this expression into (3.4). As indicated in (3.4), there are other limits that are dominated by large $\theta$. One such limit is $\log(z/\bar{z}) \to -\infty$. In this case, expanding the $\mathrm{Li}_2$ expression again leads to the same right hand side of (3.4). The result blows up, and we trust its divergent part. To find the finite part, a more careful analysis is needed, as discussed in Section 3.2. From a thermodynamic Bethe ansatz point of view, these manipulations go by the name of high-temperature analysis, dominated by very energetic particles with large rapidities.

---

## 3.1 OPE Limits

Armed with (3.3) and (3.4), we can now straightforwardly explore all the various interesting OPE limits summarized in Figure 5. They can be Euclidean OPE's (if two points approach each other), light-like OPE's (if two points become null separated) or double-light-like OPE limits (if the four points approach the cusps of a null square). For each of these limits, it matters whether adjacent operators (at points $x_i$, $x_{i+1}$) or diagonally opposite operators (at

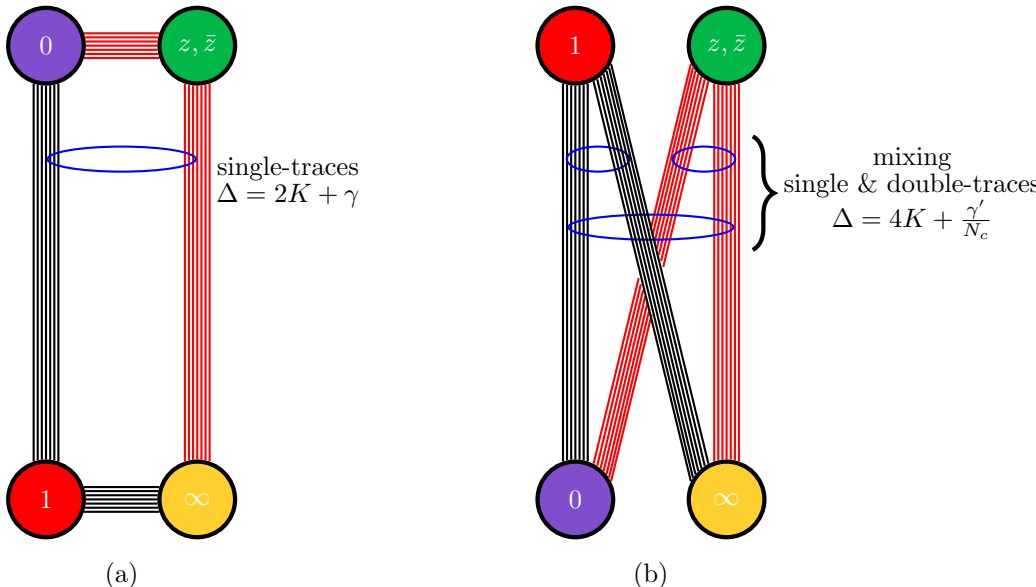

**Figure 6:** (a) The OPE $z, \bar{z} \to \infty$ or $z, \bar{z} \to 0$ is controlled by large single trace operators. (b) The other OPE limit connecting diagonals of the square, that is $z, \bar{z} \to 1$ is dominated by double traces. The area will indeed behave strikingly differently in both limits.

points $x_i$, $x_{i+2}$) collide (or become null separated). This makes a big difference, since in our correlator, operators at points $x_i$ and $x_{i+1}$ are connected by an edge of the large R-charge frame (see Figure 1), whereas operators at $x_i$ and $x_{i+2}$ are located at non-neighboring cusps of the square. This important difference is illustrated in an example in Figure 6. Hence all in all, there are six different interesting limits that we can take, as summarized in Figure 5.[8]

We find that the area $\mathbb{A} \equiv -(\log \mathbb{O})/\sqrt{\lambda}$ in the various limits is approximately given by the expressions in Table 1. A few comments are in order:

- Some areas are very large (identified with $\star$), some are very small (identified with $\ast$), and some can be either. The large areas are the ones where the points colliding or becoming null separated are neighbors in the square, while the vanishingly small area are those where the points colliding or becoming null separated are non-neighboring cusps in the square. This is in nice agreement with the intuition that the first is a single-trace OPE channel while the latter ought to be dominated by double-trace operators, see Figure 6. In particular, when the area is very large, we can extract the effective spin and dimension of the exchanged operators, since those dominate the OPE saddle point. When the area vanishes, instead, the prefactor multiplying the classical area would become important; it would be very interesting to study this prefactor, at least in the limits considered here. The technology to probe this fascinating limit should be very close to that recently developed in [11][9]

- Some limits arise as particular cases of (3.3); others of (3.4). For example, the first line is a specialization of (3.3) while the last line follows from (3.4).

---

[8]All these limits can be approached from the Euclidean regime; genuinely Lorentzian configurations will be considered in the next subsection.

[9]We thank Benjamin Basso for enlightening discussions on the physics of this OPE channel and its relation to (the singularities of) extremal three-point functions way before their paper was published.

|  | $z \to 0$ | $z \to 1$ |
|---|---|---|
| $\bar{z} \to 0$ | Euclidean OPE $\star$ <br> $4\pi^{3/2}\mathbb{A} \simeq \zeta(3/2)\sqrt{-\log(z\bar{z})}$ | Equivalent to $z \to 0, \bar{z} \to 1$ |
| $\bar{z}$ finite | Light-like OPE $\star$ <br> $4\pi^{3/2}\mathbb{A} \simeq \sqrt{\log\frac{1}{z}}\left(\zeta\left(\frac{3}{2}\right) - \mathrm{Li}_{\frac{3}{2}}(\bar{z})\right)$ | Light-like OPE $*$ <br> $2\pi\mathbb{A} \simeq \sqrt{1-z}\sqrt{-\log\bar{z}}$ |
| $\bar{z} \to 1$ | Double Light-like OPE <br> $2\pi\mathbb{A} \simeq \sqrt{1-\bar{z}}\sqrt{-\log z}$ | Euclidean OPE $*$ <br> $2\pi\mathbb{A} \simeq \sqrt{1-z}\sqrt{1-\bar{z}}$ |
| $\bar{z} \to \infty$ | Double Light-like OPE $\star$ <br> $16\pi^2\mathbb{A} \simeq (\log(-z) + \log(-1/\bar{z}))^2$ | Equivalent to $z \to 1, \bar{z} \to 0$ |

**Table 1:** The area $\mathbb{A} \equiv -(\log\mathbb{O})/\sqrt{\lambda}$ in various OPE limits.

- Most limits commute with each other. For example, from the second line we can further specialize to the elements in the first and third lines. One exception is the limit where $z \to 0$ and $\bar{z} \to \infty$ with their product held fixed. This leads to the fourth line, which is not a trivial expansion of the second line at large $\bar{z}$. In this case, the order of limits matters. This last limit, with $z \to 0$ and $\bar{z} \to \infty$, was very important in bootstrapping the octagon to all loops in [3], and will be the subject of a more detailed discussion in the next section.

- The Euclidean OPE limit $z, \bar{z} \to 1$ corresponds to $\varphi \to 0$ and $\phi \to 0$. We can reach it starting with $z \to 1$, i.e. under the conditions of (3.3), and then taking the limit $\bar{z} \to 1$, leading to the expression in Table 1, see also (F.12) in Appendix F.1. We can also first take $\varphi \to 0$ (i.e. $\bar{z} \to 1/z$) and then send $z \to 1$ to obtain a specialization of the result in the table, namely $2\pi\mathbb{A} \simeq 1 - z$. To obtain this result, we note that when $\varphi \to 0$, we are under the conditions of (3.4) even before taking any limit on $\phi$. We hence obtain, when $\varphi \to 0$,

$$\log\mathbb{O} \simeq -\frac{\sqrt{\lambda}}{4\pi^2}\,\widetilde{\phi}\,(2\pi - \widetilde{\phi}). \tag{3.9}$$

Here $\widetilde{\phi} = \phi \mod 2\pi$, since the derivation implicitly assumes that $\phi = -i/2\log(z/\bar{z}) = -i\log(z)$ is defined to be between 0 and $2\pi$ on the principal sheet. Outside this range, the result should be continued periodically to match with the manifestly periodic function (3.1) in the Euclidean regime, see Figure 4. When $\phi \to 0$, we recover indeed $2\pi\mathbb{A} \propto 1 - z$, which is consistent with the above.

- Another very important limit for the weak-coupling bootstrap of [3] was the observation of Steinmann-like relations: The expansion around $z = \bar{z} = 1$ generates only single logarithms of $z - 1$ and $\bar{z} - 1$ in perturbation theory. As such, the double discontinuity, around $z = 1$ say, vanishes. We cannot see this effect at strong coupling, since it would require knowing the one-loop pre-factor to our classical result.

## 3.2 The Null Octagon (Results and Speculations)

In this section, we would like to open a longish parenthesis and study the very interesting limit where the four points approach the cusps of a null square. More precisely, we want to

focus on the cyclic order where the points which are becoming consecutively null are those which are also connected by geodesics. In terms of cross ratios, this corresponds to the limit

$$z \to 0^-, \qquad \bar{z} \to -\infty \quad \text{with fixed } z\bar{z}. \tag{3.10}$$

This limit, also dubbed as double light-like limit, was studied in detail for small operators in perturbation theory in [12]. There, it was highlighted that this limit is controlled by the exchange of large-spin operators of leading twist. In the analysis of [12], there is a single lowest-twist family (of twist 2), which simplified the analysis and allowed for all-loop re-summations of the four-point functions (and associated OPE data) in this limit. In our case, we are dealing with large operators, so that even at leading twist there is a huge degeneracy involved. Some progress has been made towards taming this degeneracy (see e. g. [13]), but for generic large operators, this problem remains unsolved. It would be very interesting to clean this up and settle many of the interesting issues raised in the discussion that follows.

This double light-like limit was identified as very important in bootstrapping the octagon in perturbation theory in [3]. In this work, it was noted that the correlator admits a beautiful exponentiated result in this limit, as

$$\log \mathbb{O} = -\frac{(\log(-z) + \log(-1/\bar{z}))^2}{8\pi^2} \Gamma(\lambda) + \frac{1}{8} C(\lambda) + \frac{\lambda}{16\pi^2} \log(z\bar{z})^2. \tag{3.11}$$

The functions $\Gamma$ and $C$ were explicitly evaluated to twenty-four loops, and a general algorithm for finding them to arbitrarily high loop order was provided. We will come back to these terms in a moment.

For now, let us stress that the last term in (3.11) is very weird, as it is one loop exact. To our knowledge, it is the only instance where the 't Hooft coupling appears explicitly in a physical observable.[10] It seems like we could define the coupling non-perturbatively as the coefficient of $\log(z\bar{z})^2$ in the double light-like limit. In perturbation theory, this is indeed the case. However, extra care is needed if we want to go to finite coupling. We would like to claim that we cannot take (3.11) as the correct double light-like limit at finite coupling. *If* it were correct, it would lead to a correlator proportional to the exponential of $\lambda$ at strong coupling, which is much larger than the classical string tension $\sqrt{\lambda}$. It would thus contradict the most basic AdS/CFT dictionary. Indeed, above we argued that the octagon exponentiates with a nice exponent proportional to $\sqrt{\lambda}$, so there is indeed no sign of such a weird term at strong coupling. We conclude that we face a clear example of an order of limits issue. In perturbation theory, even if we take $\log(z/\bar{z})$ and $\log(z\bar{z})$ to infinity, the products $\Gamma \log^2(z/\bar{z})$ and $\lambda \log^2(z\bar{z})$ are always very small, since the coupling is the smallest parameter. Hence it is dangerous to extrapolate these expressions to finite or strong coupling, where these products would become very large.

Instead, we conjecture that in the double light-like limit (3.10) and at finite coupling, we have

$$\log \mathbb{O} \simeq -\frac{(\log(-z) + \log(-1/\bar{z}))^2}{8\pi^2} \Gamma(\lambda) + \frac{1}{8} C(\lambda) + \log \log(\bar{z}/z) f(z\bar{z}, \lambda) + h(z\bar{z}, \lambda). \tag{3.12}$$

---

[10]Unless we count the magnon dispersion relation as a physical quantity. Strictly speaking, we measure anomalous dimensions and not the magnon dispersion, and hence it is not a direct observable.

Strong evidence for this proposal comes from the strong-coupling analysis performed here. We observe precisely this structure, with

$$\Gamma \simeq \frac{\sqrt{\lambda}}{2}, \qquad C \simeq -2\sqrt{\lambda}, \qquad f \simeq \frac{\sqrt{\lambda}}{16\pi^2} \log(z\bar{z})^2. \tag{3.13}$$

The subleading contribution $h$ is considerably more complicated. It vanishes if $z = 1/\bar{z}$, and admits a simple series expansion away from this limit, see Appendix F.2. Inspired by the type of expressions observed for small operators (see below), it is tempting to interpret the function $f$ as a sort of recoil contribution, and conjecture that $f \propto \Gamma(\lambda) \log(z\bar{z})^2$ in the strict light-like limit – i.e. when the coupling is *not* the smallest parameter.

In a beautiful recent work [14], Belitsky and Korchemsky addressed the double light-like limit of the octagon in the further simplifying *diagonal limit*

$$z = 1/\bar{z}. \tag{3.14}$$

This is indeed a beautiful simplifying limit, which projects out the last subtle term, leading to

$$\log \mathbb{O}(z, 1/z) = -\frac{\log(-z)^2}{2\pi^2} \Gamma(\lambda) + \frac{1}{8} C(\lambda) \qquad \text{for } z \to 0. \tag{3.15}$$

The authors of [14] found that the octagon determinant representation of [4, 15] simplifies enormously in this diagonal light-like limit, and took advantage of this simplification to derive analytic expressions for both $\Gamma$ and $C$ as

$$\Gamma_{BK}(\lambda) = \log\cosh\frac{\sqrt{\lambda}}{2}, \qquad C_{BK}(\lambda) = -\log\frac{\sinh\sqrt{\lambda}}{\sqrt{\lambda}}. \tag{3.16}$$

They observed that these re-summed expressions have a rather remarkable property: Their strong-coupling expansions truncate at one-loop order! Namely,

$$\Gamma_{BK}(\lambda) = \frac{\sqrt{\lambda}}{2} - \log 2 - \sum_{n=1}^{\infty} \frac{(-1)^n}{n} e^{-n\sqrt{\lambda}}, \quad C_{BK}(\lambda) = -\sqrt{\lambda} + \frac{1}{2}\log\lambda + \log 2 + \sum_{n=1}^{\infty} \frac{e^{-2n\sqrt{\lambda}}}{n}. \tag{3.17}$$

It would be fascinating to understand why these series truncate. Note also that $C(\lambda)$ has an amusing logarithmic term, which seems to indicate an interesting prefactor structure of the octagon in this double light-like limit as

$$\mathbb{O} \simeq e^{-\frac{\sqrt{\lambda}}{4\pi^2}\log(-z)^2 - \frac{\sqrt{\lambda}}{8}} \lambda^{1/16} \times O(1), \tag{3.18}$$

which would also be fascinating to understand. Now, the attentive reader probably noticed that the classical piece in $\Gamma$ in (3.17) perfectly agrees with our strong coupling evaluation, while the classical part in $C$ is off by a factor of two.

What is this factor of two? One option is a glitch in our computation.[11] The other option is physics. Perhaps both results are perfectly correct, and the disagreement is simply because

---

[11] We carefully checked our computation, and all factors appear to be correct. At strong coupling, the limit when $z \to 1/\bar{z}$ is a simple high-temperature limit of the free-energy type formula for the area, for which we have good control over $C(\lambda)$. Also, it is hard to see how a potentially missing factor of two in $C(\lambda)$ of [14] would come about, since we did check that the expression for $C(\lambda)$ in [14] does agree with the perturbative results of [3] up to 24 loops.

we are again taking different limits. In this work, we first go to strong coupling $\lambda \to \infty$, and then take the double light-like limit (3.10). In [14] on the other hand, the light-like limit is taken first. So our result could be a purely classical/minimal area result, while the prediction from [14] appears to be a more strict double light-like limit.

There was a similar setup in the context of null polygonal Wilson loops, where such limits were similarly subtle. In [16], it was understood that in the collinear limit of scattering amplitudes (corresponding to an OPE limit of Null Wilson loops [17]), an additional enhanced contribution from nearly massless scalars modifies the classical minimal area result by an additional constant term, and also produces a funny power of $\lambda$ in the prefactor. Could the mismatch we are observing, together with the interesting prefactor in (3.18), have a similar origin? Perhaps in the double light-like limit additional massless modes come into play, the naive expansion around the BMN vacuum needs to be reorganized, and a more careful analysis is needed along those lines? We are currently analyzing this possibility.

It is very likely that this potential order of limits problem and the order of limits issue related to the function $f$ introduced above are not mathematically unrelated. So another source of clarification would come from repeating the double light-like OPE limit analysis of [14] without the diagonal restriction (3.14) to see if the picture above – including the interesting function $f$ – is indeed realized. Expanding around the diagonal limit, that is for $\log(z\bar{z})^2 \ll 1$ should hopefully not be that hard, and would be very illuminating.

Finally, we could compare the discussion above with the double light-like predictions for small operators from Alday and Bissi in [12, 13] which we alluded to above. We use the notation of [14], equation (1.10) therein, from where we extract the correlator in the null limit as

$$
G_4 \simeq H(g) \int_0^\infty \left( \prod_{j=1}^2 2K_0(2\sqrt{y_j})dy_j \right) e^{-S}, \quad S = \tfrac{1}{2}\Gamma_{\text{cusp}} \log\left(-\tfrac{z}{y_1}\right) \log\left(-\tfrac{1}{\bar{z}y_2}\right) - \tfrac{1}{2}\Gamma_v \log\left(\tfrac{z}{\bar{z}y_1 y_2}\right),
$$
(3.19)

where $K_0$ is the modified Bessel function of the second kind, and the contribution shown explicitly is what is interpreted as the recoil contribution. In the null limit, we can estimate the integral by saddle point,[12] so that finally

$$
\begin{aligned}
\log(G_4) \simeq &-\tfrac{1}{2}\Gamma_{\text{cusp}}\log(-z)\log(-1/\bar{z}) + \tfrac{1}{2}\gamma \log(z/\bar{z}) \\
&+ \Gamma_{\text{cusp}}\log(-1/\bar{z})\log\left(\Gamma_{\text{cusp}}\log(-1/\bar{z})\right) + \Gamma_{\text{cusp}}\log(-z)\log\left(\Gamma_{\text{cusp}}\log(-z)\right) \\
&+ \text{constant},
\end{aligned}
$$
(3.20)

where $\gamma = \Gamma_v - \log(4e^2)\Gamma_{\text{cusp}}$. This expression is indeed quite similar to the expressions above. However, the $\log\log(-z)$ term here is dressed by a term $\log(-z)$ that is linearly divergent in the double light-like limit, while for us it was multiplied by a finite factor $\log(z\bar{z})^2$, which is held fixed in the limit (3.10). The second term in the first line is also absent for us. Perhaps this is related to the absence of recoil in our correlator, with its huge R-charge frame. It would be fascinating to investigate this further, i.e. to analyze the octagon from a light-cone bootstrap perspective.

In sum, we have a very rich behavior of the correlation function given by the octagon in the null limit.[13] We can approach it in three different ways:

---

[12]The leading saddle location is at $(y_1, y_2) \simeq \Gamma_{\text{cusp}}^2 \left(\log(-z)^2, \log(-1/\bar{z})^2\right)/4$.

[13]We thank Grisha Korchemsky and Andrei Belitsky for useful correspondence on these matters.

- Weak coupling: We first expand around $\lambda = 0$, and then expand in the light-cone limit (3.10). This was originally studied in [3] to all loop orders in perturbation theory.

- Null Limit First: We first take the light-cone limit (3.10), keeping $z\bar{z}$ fixed. In its generic form, this limit was not analyzed yet. In the diagonal limit, where $z\bar{z} = 1$, it was recently studied in [14].

- Strong coupling: We first take $\lambda \to \infty$, and then expand in the light-cone limit (3.10). This is what we studied in this section.

All three limits lead to beautiful exponentiation of the correlator, as just reviewed. The leading term in the exponent seems to be universal, and independent on how we approach the null limit. It is governed by the function $\Gamma_{\text{cusp}}$. The subleading terms, however, seem sensitive to the order of limits. In particular, in the diagonal limit (which as of now is the only one we can compare), the second and third limits differ in a mild way, by a factor of two in the constant subleading term. We suspect this 2 to be a smoking gun for some yet to be unveiled interesting physics.[14]

## 3.3 Lorentzian Continuations

Lorentzian correlators can be obtained as analytic continuations of the Euclidean correlator. This follows from a Wick rotation of the time-coordinates of the operators, whose effect on the cross ratios is to take them from being complex conjugate $\bar{z} = z^*$ in the original Euclidean configuration to real and independent values in the final Lorentzian configuration. If we start in the Euclidean cylinder $\mathbb{R} \times S^3$, then at the end of the Wick rotation we are in the Lorentzian cylinder, where $\mathbb{R}$ is a time. We will only use a circle subspace of the full three-sphere, so we will be working in an $\mathbb{R} \times S^1$ subspace. Then each operator insertion is parametrized by an angle $\phi_j$ on the circle and a time $t_j - i\epsilon_j$, where the order of the imaginary epsilons dictates the order of the operator insertions, see e.g. [18]. The physical cross-ratios then take the nice form

$$z = \frac{\sin\psi_{12}^+ \sin\psi_{34}^+}{\sin\psi_{13}^+ \sin\psi_{24}^+} \qquad \Longleftrightarrow \qquad 1 - z = \frac{\sin\psi_{14}^+ \sin\psi_{23}^+}{\sin\psi_{13}^+ \sin\psi_{24}^+} , \qquad (3.21)$$

where $\psi_{ij}^+ = \psi_i^+ - \psi_j^+$, with $\psi_j^+ = (\phi_j + t_j)/2$, and with a similar expression for $\bar{z}$ in terms of the other light-cone direction $\psi_j^+ \to \psi_j^- = (\phi_j - t_j)/2$.

Consider the setup where operators $O_1$ and $O_3$ are lifted in Lorentzian time and enter the light-cone(s) of the other pair of operators $O_2$ and $O_4$, see figure Figure 7. As indicated in the figure, we move up the Lorentzian cylinder along the light-cone directions $\psi^+$, so that nothing relevant happens along the $\psi^-$ directions; we can thus ignore the $\bar{z}$ cross-ratio altogether. We do not touch operators $O_2$ and $O_4$, and we move $O_1$ and $O_3$ simultaneously, so that their distance is always space-like; then we can also ignore the denominator factors in (3.21) which always stay finite, basically untouched. All the fun is in the four sine factors in the numerators in (3.21). To figure out what happens there, we need the $i\epsilon$'s. Since we are moving $O_1$ and $O_3$ up, and are looking for a time-ordered configuration at the very end,

---

[14]The extrapolation from weak coupling differs from the strong-coupling result in an even more drastic fashion in the non-diagonal limit. But in that case, we have no reason to expect the difference to be explained by some interesting physics, since the two limits are genuinely very different. In the weak-coupling limit, $\lambda^n \log(z/\bar{z})$ is very small; in the second and third limits, this combination is very large.

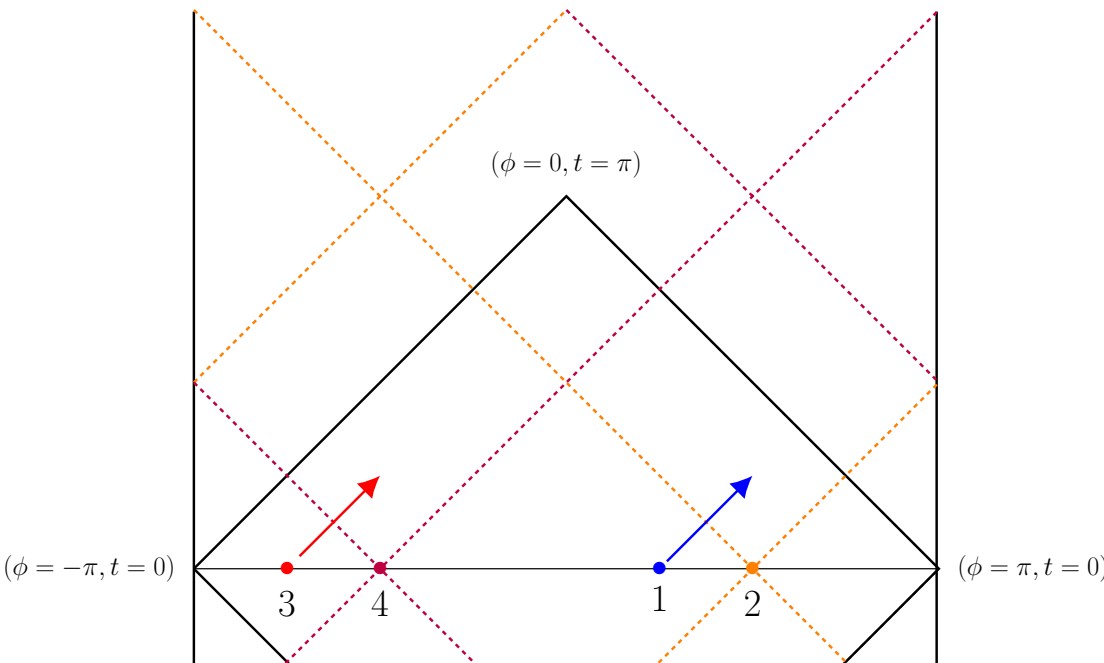

**Figure 7:** Operators $O_4$ and $O_1$ being lifted in Lorentzian time $t$, entering the light-cones of $O_1$ and $O_3$ as we perform the Wick rotation. As reference, we depict a Poincare patch as a solid diamond, specifying its corners in global coordinates $t$ and $\phi$ of $R_t \times S_\phi$. The lines $\phi = -\pi$ and $\phi = \pi$ must be identified.

it suffices to take $\epsilon_1 = \epsilon_3 > 0 = \epsilon_2 = \epsilon_4$. Then the numerators never vanish. Instead, each of them picks an $e^{i\pi}$ half-monodromy as $O_1$ or $O_3$ crosses a light-cone of $O_2$ or $O_4$. Two such half-monodromies can then combine into full monodromies. For example, when $O_1$ crosses the light-cone of $O_2$, and $O_3$ crosses the light-cone of $O_4$, then each factor in $z$ picks a half-monodromy, so that $z$ acquires a full monodromy around zero, as illustrated in Figure 8. If we continue moving up the Lorentzian cylinder, two more light-cones are crossed, and then it will be $1 - z$ numerator factors which become relevant, and we now end up picking an extra monodromy, this time around $z = 1$, see also Figure 8.

This describes the analytic continuations required to move up the Lorentzian cylinder along the light-like helices, where $z$ changes and picks monodromies, whereas $\bar{z}$ does not. What about other paths? They all give the same of course. For example, paths using the other light-cone helices would generate $\bar{z}$ monodromies instead, with counter clock-wise orientation. Since we start in the Euclidean correlator, which is single-valued, we can always trade those monodromies for regular clock-wise oriented $z$ monodromies, see e.g. [19]. Similarly, if we were to move the points vertically, we would generate both types of monodromies, which we could again relate to purely $z$ monodromies using single-valuedness, so that all is nice and consistent.

What about moving other points up the Lorentzian cylinder? That would of course be different, but trivially related to the previous case, since we can reach these other cases by simply relabeling the points. For instance, with the exchange $O_3 \leftrightarrow O_4$, we have $z \to z/(z-1)$, and the corresponding time-ordered correlator is obtained by the monodromies

$$\mathbb{O}^2 \xrightarrow[\text{LCs of } O_2/O_3]{\overset{O_1/O_4 \text{ cross}}{}} \mathcal{C}_0 \circ \mathbb{O}^2 \xrightarrow[\text{LCs of } O_3/O_2]{\overset{O_1/O_4 \text{ cross}}{}} \mathcal{C}_\infty \circ \mathcal{C}_0 \circ \mathbb{O}^2 \xrightarrow[\text{LCs of } O_2/O_3]{\overset{O_1/O_4 \text{ cross}}{}} \mathcal{C}_0 \circ \mathcal{C}_\infty \circ \mathcal{C}_0 \circ \mathbb{O}^2 \ldots$$

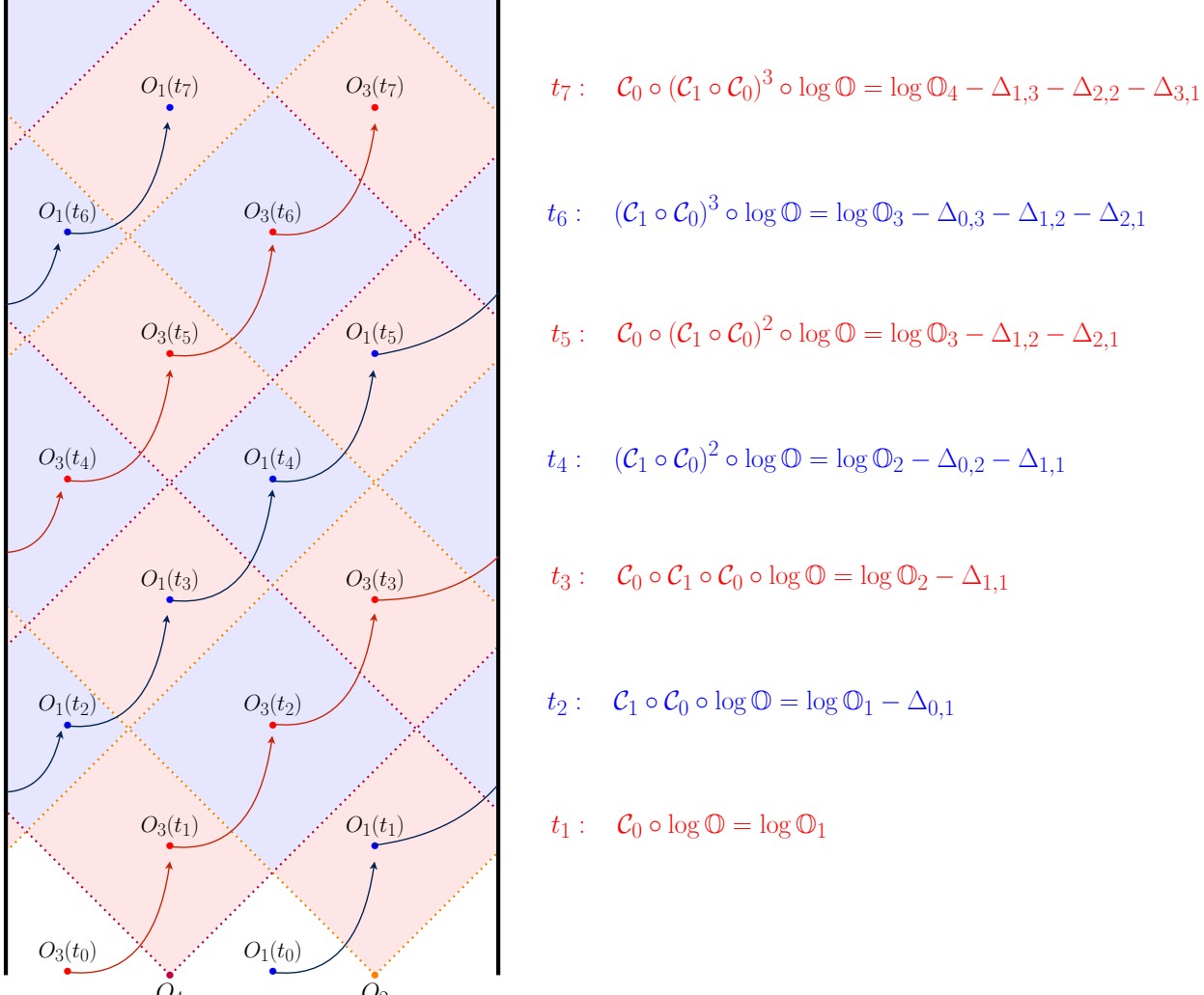

$$t_7: \quad \mathcal{C}_0 \circ (\mathcal{C}_1 \circ \mathcal{C}_0)^3 \circ \log \mathbb{O} = \log \mathbb{O}_4 - \Delta_{1,3} - \Delta_{2,2} - \Delta_{3,1}$$

$$t_6: \quad (\mathcal{C}_1 \circ \mathcal{C}_0)^3 \circ \log \mathbb{O} = \log \mathbb{O}_3 - \Delta_{0,3} - \Delta_{1,2} - \Delta_{2,1}$$

$$t_5: \quad \mathcal{C}_0 \circ (\mathcal{C}_1 \circ \mathcal{C}_0)^2 \circ \log \mathbb{O} = \log \mathbb{O}_3 - \Delta_{1,2} - \Delta_{2,1}$$

$$t_4: \quad (\mathcal{C}_1 \circ \mathcal{C}_0)^2 \circ \log \mathbb{O} = \log \mathbb{O}_2 - \Delta_{0,2} - \Delta_{1,1}$$

$$t_3: \quad \mathcal{C}_0 \circ \mathcal{C}_1 \circ \mathcal{C}_0 \circ \log \mathbb{O} = \log \mathbb{O}_2 - \Delta_{1,1}$$

$$t_2: \quad \mathcal{C}_1 \circ \mathcal{C}_0 \circ \log \mathbb{O} = \log \mathbb{O}_1 - \Delta_{0,1}$$

$$t_1: \quad \mathcal{C}_0 \circ \log \mathbb{O} = \log \mathbb{O}_1$$

**Figure 8:** Left panel: Lorentzian configurations with operators $O_3$ and $O_1$ inside the light-cones of $O_4$ and $O_2$, at different Lorentzian times $t_j$ in $\mathbb{R}_t \times S^1$. Right panel: the correspondent monodromies $\mathcal{C}_x$, counter-clockwise around the branch points $x = 0$ or $1$, and their effect on the Euclidean correlator $\log \mathbb{O}$. See (3.23) and (3.24) for notation used on the right-hand side.

$$= \frac{\langle 0|O_4(t - i\epsilon, \vec{x}_4)O_1(t - i\epsilon, \vec{x}_1)O_3(0, \vec{x}_3)O_2(0, \vec{x}_2)|0\rangle}{\texttt{Tree Level}} \quad (3.22)$$

once we move $O_4$ and $O_1$ up the cylinder. In practice, it is useful to replace the infinity monodromy by counterclockwise monodromies around 1 and 0, as $\mathcal{C}_\infty = \bar{\mathcal{C}}_1 \circ \bar{\mathcal{C}}_0$.

This concludes the analysis of the analytic continuation paths, that is of the kinematics. For the dynamics, we would like to know how the octagon transforms under all such monodromies as we analytically continue it from its Euclidean representation (3.1) into Lorentzian kinematics. There are two important contributions. On the one hand, we have the variables $\varphi$ and $\phi$, which are themselves *not* periodic as we perform monodromies around $z = 0$. Instead, they transform as $(\varphi, \phi) \to (\varphi - i\pi n, \phi + \pi n)$ under $n$ monodromies around $z = 0$. The area (3.1) is not periodic in such shifts,[15] hence we get an obvious contribution,

---

[15]The function $\log \mathbb{O}$ is $2\pi n$ periodic in $\phi$, but has no obvious periodicity in $\varphi$.

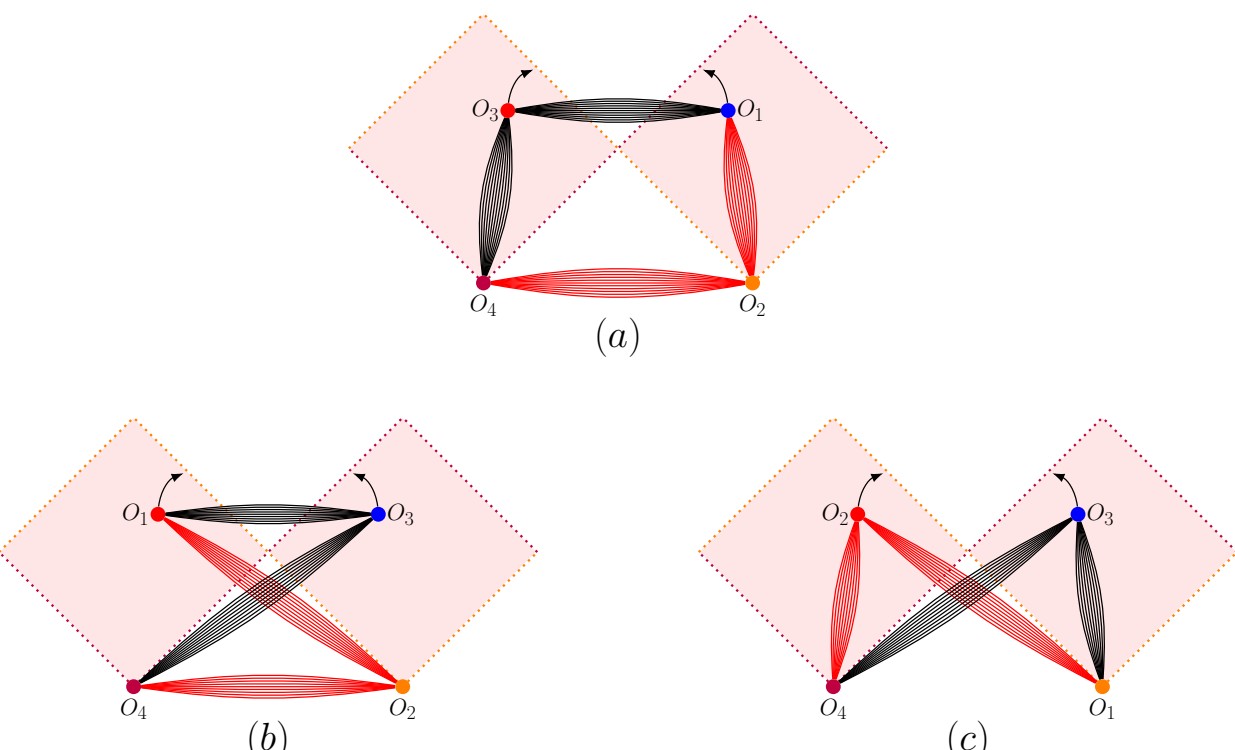

**Figure 9:** The large R-charges in our setup admit three qualitatively different Regge limits, which are reached as follows: *(a)* $z, \bar{z} \to 1$ after $z \circlearrowleft 0$, *(b)* $z, \bar{z} \to 0$ after $z \circlearrowleft 1$, *(c)* $z, \bar{z} \to 0$ after $z \circlearrowleft \infty$.

which we conveniently define as

$$\log \mathbb{O}_n \equiv \log \mathbb{O}(\varphi - i\pi n, \phi + \pi n) \,, \tag{3.23}$$

This is however not the full story. As we analytically continue the cross-ratios, the singularities of the integrand will move, and can cross the contour of integration, at which point we produce further interesting contributions. This does indeed happen, albeit not for the $z = 0$ monodromies; it happens for the monodromies around $z = 1$. Such extra new terms are very simple and take the form

$$\Delta_{p,q} := \frac{\sqrt{\lambda}}{\pi} \sqrt{\Big(\log(z) + 2\pi i p\Big)\Big(\log(\bar{z}) + 2\pi i q\Big)} \,. \tag{3.24}$$

All octagon expressions in any Lorentzian region can be cast as simple combinations of $\log \mathbb{O}_n$ and $\Delta_{p,q}$, as summarized in the example of Figure 8. In concise formulae, carefully derived in Appendix D,

$$(\mathcal{C}_1 \circ \mathcal{C}_0)^n \circ \log \mathbb{O} = \log \mathbb{O}_n + \sum_{i=0}^{n-1} \Delta_{i,n-i} \,,$$
$$\mathcal{C}_0 \circ (\mathcal{C}_1 \circ \mathcal{C}_0)^n \circ \log \mathbb{O} = \log \mathbb{O}_{n+1} + \sum_{i=0}^{n-1} \Delta_{i+1,n-i} \,. \tag{3.25}$$

As a simple application, we can consider the Regge limit depicted in Figure 9. As depicted there, since our configuration is carries large R-charges, the behavior in this limit can be

strikingly different, depending on whether the dominating exchange is charged or chargeless. Indeed, we see that the area can either blow up or vanish, depending on the R-charge setup! One of these Regge configurations can be reached by taking a monodromy around $z = 0$ and then approaching $z, \bar{z} = 1$, and this limit is dominated by double-trace exchanges, see Figure 9a. For the area function $\log \mathbb{O}$, we find in this case

$$z, \bar{z} \to 1 \text{ after } z \circlearrowleft 0 \; : \qquad \log \mathbb{O} \simeq -\frac{\sqrt{\lambda}}{2\sqrt{\pi}}(1-i)(\sqrt{1-z} + \sqrt{1-\bar{z}} - \sqrt{2-z-\bar{z}}) \, . \quad (3.26)$$

The other configurations are dominated by single-trace exchange, they are reached by taking $z$ around 1 (or $\infty$) and then approaching $z, \bar{z} = 0$, as represented in Figure 9b(c). In this case, we find for the area function:[16]

$$z, \bar{z} \to 0 \text{ after } z \circlearrowleft 1 \text{ or } z \circlearrowleft \infty \; : \qquad \log \mathbb{O} \simeq +\frac{\sqrt{\lambda}}{\pi}\sqrt{\log(z)\log(\bar{z})} \, . \quad (3.27)$$

We could have obtained (3.27) directly from the light-like OPE expression in the second row of Table 1 in the previous section ($z \to 0$, $\bar{z}$ finite). For the limit (3.26), we need to be more careful, since we want to approach $z, \bar{z} = 1$ with a fixed angle. In this case, we can obtain the expression directly from the expansion in Appendix F.1. With hindsight, the attentive reader could argue that with these two shortcuts, we could have sidelined all the subtle continuations and contour analysis for this particular Regge limit. That is true. Our apologies to the reader.

# 4 Conclusions

The main result of this short note is a compact representation for the octagon at strong coupling. It takes the form

$$\mathbb{O} \simeq e^{-\sqrt{\lambda}\,\mathbb{A}(z,\bar{z})} \, . \quad (4.1)$$

It would be very interesting to compute the function $\mathbb{A}$ directly from string theory. It should be a nice minimal area. It would be even more interesting to compute the one-loop prefactor multiplying the exponential, both from the integrability representation for the octagon and from string theory. Together with the area, the prefactor should provide strong insights about the finite-coupling nature of this object. We also expect many interesting properties of the correlator, such as the bulk point limit and various Steinmann discontinuity properties to be manifest only once we tackle this prefactor.

For physical kinematics (both Euclidean and Lorentzian), the real part of $\mathbb{A}$ is positive. This is probably good, since otherwise we would obtain an exponentially large octagon. As it stands, we obtain an exponentially small one, as expected for a tunneling process. A particular implication is that in the non-planar limit explored in [2], the correlator becomes simply given by the very simple expression (4.3) therein.

On the other hand, we can easily take unphysical kinematics and obtain an exponentially large correlator. For example, we can take (3.4) in the unphysical regime where $\phi =$

---

[16]Note that the octagon becomes exponentially large in this Lorentzian Regge limit (i.e. the area becomes negative). A similar correlator in an equivalent conformal Regge limit was considered in [20], see Section 4 there. In that context, the correlator serves as a measure of chaotic behavior, and also displays a positive (Lyapunov) exponent. It would be interesting to better understand the meaning of our result (3.27) in the context of [20].

$(1/2i)\log(z/\bar{z})$ is real and very large to get a huge octagon. This is not surprising. It happens commonly in such classical analyses, as in the high-energy scattering of strings in flat space by Gross–Mende [21] and Gross–Manes [22], as recently highlighted by Sever and Zhiboedov [23]. There, when $s/t$ is very large and $t$ is negative, we are in a physical regime, and the amplitude is exponentially small indeed, but if $t$ is positive, then we get an exponentially large result. In these complexified situations, where the octagon is very large, we can again take advantage of the non-planar re-summations in [2] to conclude that the correlator would now be given by (4.4) therein. It would be very interesting if there was some universality in these large-area limits, akin to those recently explored by [24] and [23] in the context of Regge theory.

It is rare to have access to a full-fledged four-point function at strong coupling. As illustrated here, the result exhibits a plethora of interesting limits and monodromies, as expected in a rich strong-coupling CFT as the one under consideration. In Appendix E, we compare these properties with those observed in the perturbative weak-coupling regime. It would be very nice to address these at finite coupling, perhaps making use of the recent determinant representation [4]. Perhaps the most intriguing limit of all – and thus the most interesting to address, given all the puzzles of Section 3.2 – would be the double light-like limit.

## Acknowledgments

We thank Benjamin Basso, Andrei Belitsky, Nathan Berkovits, João Caetano, Simon Caron-Huot, Thiago Fleury, Valentina Forini, Vasco Gonçalves, Andrea Guerrieri, Grisha Korchemsky, Ivan Kostov, Juan Maldacena, João Penedones, Amit Sever, Sasha Zhiboedov and Shota Komatsu for numerous enlightening discussions and suggestions. We are specially grateful to Vasco Gonçalves for collaboration at early stages of this work. Research at the Perimeter Institute is supported in part by the Government of Canada through NSERC and by the Province of Ontario through MRI. This work was additionally supported by a grant from the Simons Foundation #488661. T. B. acknowledges support from DESY (Hamburg, Germany), a member of the Helmholtz Association HGF. The work of T. B. was funded by the Deutsche Forschungsgemeinschaft (DFG, German Research Foundation) – 460391856.

## A    Octagon Correlators

It was explained in [1] and [2] how the octagon $\mathbb{O}_{\ell=0}$ with zero bridge length completely captures the planar and non-planar loop corrections of a specific "simplest" correlator. We will briefly review (a slight variation of) the argument in the following, and thereby justify the identification $\alpha = \bar{\alpha} = 1$ in (2.4). For more details, see Sections 4.1 and 4.2 in [1].

The cleanest isolation of the octagon (with zero internal bridge length) occurs for a correlator of the four operators

$$\mathcal{O}_1 = \text{tr}(\bar{Z}^k \bar{X}^k) + (\text{permutations}), \qquad \mathcal{O}_2 = \text{tr}(X^{2k}),$$
$$\mathcal{O}_3 = \text{tr}(Z^{2k}), \qquad\qquad\qquad\qquad \mathcal{O}_4 = \text{tr}(\bar{Z}^k \bar{X}^k) + (\text{permutations}) \qquad (A.1)$$

at large $k$ in the planar limit. Here, $X$ and $Z$ are orthogonal complex scalars, e.g. $X = \phi_1 + i\phi_2$ and $Z = \phi_5 + i\phi_6$. The operators $\mathcal{O}_2$ and $\mathcal{O}_3$ are BPS superconformal primaries, and $\mathcal{O}_1$ and

$\mathcal{O}_4$ are BPS descendants. At tree level, the correlator $\langle\mathcal{O}_1\ldots\mathcal{O}_4\rangle$ is given by a single square-shaped Feynman diagram, where each edge of the square consists of $k$ parallel propagators. In the large-$k$ limit, all loop corrections are confined to the individual regions "inside" and "outside" the square tree-level graph.

In the hexagonalization prescription [9], all loop corrections are captured by hexagon form factors. Two such hexagon form factors fuse to an octagon, and there is one octagon inside and one octagon outside the square tree-level graph. However, two of the physical edges of each octagon touch the descendant operators $\mathcal{O}_1$ and $\mathcal{O}_4$. These descendants carry a large number $\sim k$ of (zero-momentum) magnons on top of a BPS primary "vacuum" (e.g. $\mathrm{tr}(\bar{Z})$ or $\mathrm{tr}(\bar{X})$). The presence of these physical magnons complicate the computation of the octagon form factor.

This complication can be avoided by considering a slightly different correlator of BPS primary operators

$$\mathcal{O}(y) := \mathrm{tr}\!\left[(y\cdot\varPhi)^{2k}\right], \qquad y^2 = 0\,, \tag{A.2}$$

with (for example)

$$y_1 = \frac{1}{\sqrt{2}}(\beta_1, -i\beta_1, 0, 0, 1, -i)\,, \qquad\qquad y_2 = \frac{1}{\sqrt{2}}(1, i, 0, 0, 0, 0)\,,$$

$$y_3 = \frac{1}{\sqrt{2}}(0, 0, 0, 0, 1, i)\,, \qquad\qquad y_4 = \frac{1}{\sqrt{2}}(1, -i, 0, 0, \beta_4, -i\beta_4)\,. \tag{A.3}$$

Here, $\varPhi = (\phi_1, \ldots, \phi_6)$ are the six real scalars of $\mathcal{N} = 4$ SYM, and $y_i$ are six-component complex null vectors that parametrizes the internal polarizations of the operators. Since all operators $\mathcal{O}(y_i)$ are BPS primary "vacua", there will be no magnons on any physical edge of the two octagons. The polarizations $y_i$ are chosen such that all tree-level graphs are still square-shaped:

$$\begin{array}{c}
m \\
\boxed{1}\!\!\overset{\text{\textcolor{red}{≣}}}{}\!\!\boxed{2} \\
2k - m \;\; \Vert \qquad\qquad \Vert \;\; 2k - m \\
\boxed{3}\!\!\overset{\text{≣}}{}\!\!\boxed{4} \\
m
\end{array} \tag{A.4}$$

but the bridge length $m$ (number of parallel propagators) may take all values $0 \le m \le 2k$ (and $2k - m$ accordingly). The hexagonalization prescription requires to sum over all these skeleton graphs with different $m$. For small (or large) values of $m$, there will be interactions between the inside and the outside of the graph (the front and back of the lower left world-sheet in Figure 2). In order to confine all interactions to the individual inside and outside octagons, we have to restrict to $m \sim k$, as is made sure by the choice of operators (A.1). Moreover, the correlator $\langle\mathcal{O}_1\ldots\mathcal{O}_4\rangle$ can be extracted from $\langle\mathcal{O}(y_1)\ldots\mathcal{O}(y_4)\rangle$ by a simple projection: The reduced correlators

$$\mathcal{G} := \left(x_{12}^2 x_{34}^2\right)^{2k}\!\left\langle\mathcal{O}_1(x_1)\ldots\mathcal{O}_4(x_4)\right\rangle, \qquad \tilde{\mathcal{G}} := \left(\frac{x_{12}^2 x_{34}^2}{y_{12}^2 y_{34}^2}\right)^{2k}\!\left\langle\mathcal{O}(y_1, x_1)\ldots\mathcal{O}(y_4, x_4)\right\rangle \tag{A.5}$$

depend only on the spacetime and internal cross ratios (2.5) and

$$\alpha\bar{\alpha} = \frac{y_{12}^2 y_{34}^2}{y_{13}^2 y_{24}^2} = \sigma\,, \qquad (1-\alpha)(1-\bar{\alpha}) = \frac{y_{14}^2 y_{23}^2}{y_{13}^2 y_{24}^2} = \tau\,, \tag{A.6}$$

and by R-symmetry conservation, $\mathcal{G}$ can be extracted from $\tilde{\mathcal{G}}$ via

$$\mathcal{G}(z, \bar{z}) = \left[\tilde{\mathcal{G}}(z, \bar{z}, \sigma, \tau)\right]_{\text{coefficient of } \tau^0 \sigma^{-k}}. \tag{A.7}$$

The correlator $\tilde{\mathcal{G}}$ is a finite power series in $\sigma$. From the hexagonalization point of view, there are two sources for powers of $\sigma$: The octagons $\mathbb{O}_{\ell=0}$ inside and outside the square skeleton graph, and the skeleton graph (A.4) itself, which is proportional to $(y_{12}^2 y_{34}^2/\sigma)^{2k}\sigma^m$. Each skeleton graph is weighted by the same function $\mathbb{O}_{\ell=0}^2$. The latter has a power expansion $\sum_{i=-p}^{p} c_i \sigma^i$, with $p \leq L$ at $L$ loops.[17] Together with the final projection to the $\sigma^{-k}$ coefficient, each skeleton graph thus picks a different term in the octagon expansion. At large enough $k$, each term is picked exactly once, and thus the whole sum amounts to the full octagon squared function $\mathbb{O}_{\ell=0}^2$ evaluated at $\sigma = 1$. The polarizations (A.3) in addition imply $\tau = 0$. Together, this is equivalent to $\alpha = \bar{\alpha} = 1$, and therefore

$$\mathcal{G} = \mathcal{G}^{\text{tree}}\,\mathbb{O}_{\ell=0}^2(\alpha = 1, \bar{\alpha} = 1). \tag{A.8}$$

For general values of $\alpha$ and $\bar{\alpha}$, the parameters (angles) of the transformation $g$ in the octagon expression (2.3) are

$$\phi = -\frac{i}{2}\log\frac{z}{\bar{z}}, \qquad \theta = -\frac{i}{2}\log\frac{\alpha}{\bar{\alpha}}, \qquad \varphi = \frac{1}{2}\log\frac{\alpha\bar{\alpha}}{z\bar{z}}. \tag{A.9}$$

In terms of these angles, the character takes the form

$$W_{\{a_i\}} = \frac{W_{\{a_i\}}^+ + W_{\{a_i\}}^-}{2}, \qquad W_{\{a_i\}}^{\pm} = \prod_{j=1}^{n} 2\Big(\cos\phi - \cosh(\varphi \pm i\theta)\Big)\frac{\sin(a_j\phi)}{\sin\phi}. \tag{A.10}$$

For $\alpha = \bar{\alpha} = 1$, the angles reduce to (2.4). In this case,

$$2\Big(\cos\phi - \cosh(\varphi \pm i\theta)\Big) = -\frac{(1-z)(1-\bar{z})}{\sqrt{z\bar{z}}} = -4\sinh\left(\frac{\varphi}{2} + \frac{i\phi}{2}\right)\sinh\left(\frac{\varphi}{2} - \frac{i\phi}{2}\right), \tag{A.11}$$

and therefore (A.10) becomes (2.7).

# B   Including a Finite Bridge Length

It is straightforward to include a finite bridge-length in the octagon strong-coupling derivation by simply adding an extra factor $e^{-a\ell/\sqrt{u^2-4g^2}}$ to $T_a$ in (2.13). This leads to a slightly less pretty expression in the integration-by-parts step (2.15), and to an $i0$ prescription in the subsequent $\theta$ shift. That is the only modification, so we now have[18]

$$\frac{1}{n}\int_{-2g}^{2g} du\, e^{in\varphi\frac{u}{\sqrt{4g^2-u^2}} - n(\mathcal{L}\equiv\ell/2g)\frac{2g}{\sqrt{4g^2-u^2}}} = \frac{2g}{n}\int_{-\infty}^{\infty} d\theta\, \frac{d\tanh(\theta)}{d\theta} e^{in\varphi\sinh(\theta) - n\mathcal{L}\cosh(\theta)}$$

---

[17]At tree level, the correlator $\langle\mathcal{O}_1\ldots\mathcal{O}_4\rangle$ is exactly given by the skeleton graph (A.4) with $m = k$. At higher loop orders, there will also be contributions from skeleton graphs with $m$ deviating from $k$ by (small) finite numbers.

[18]Here we assumed $\varphi < 0$ – the opposite of the main text – to illustrate how the shift would go in that case. For the relation between positive and negative $\varphi$ and on the symmetry which flips the sign of $\varphi$, see Appendix D.1.

$$= -2gi \int_{-\infty}^{\infty} d\theta \left( \varphi \sinh(\theta) + i\mathcal{L} \frac{\sinh^2(\theta)}{\cosh(\theta)} \right) e^{in\varphi \sinh(\theta) - n\mathcal{L}\cosh(\theta)}$$

$$= 2g \int_{-\infty}^{\infty} d\theta \left( \varphi \cosh(\theta) + i\mathcal{L} \frac{\cosh^2(\theta)}{\sinh(\theta + i0)} \right) e^{n\varphi \cosh(\theta) + in\mathcal{L}\sinh(\theta)} \qquad \text{(B.1)}$$

when getting rid of the $1/n$ obstruction to factorization, and thus find

$$\log \mathbb{O}_l \simeq \frac{\sqrt{\lambda}}{2\pi} \int_{-\infty}^{\infty} \frac{d\theta}{2\pi} \left( \varphi \cosh(\theta) + i\mathcal{L} \frac{\cosh^2(\theta)}{\sinh(\theta + i0)} \right) \log(1 + Y_\ell(\theta)), \qquad \text{(B.2)}$$

where $Y_\ell(\theta)$ is given by (2.22) with $\varphi \cosh(\theta) \to \varphi \cosh(\theta) + i\mathcal{L} \sinh(\theta)$, with $\mathcal{L} \equiv \ell/2g$. If the bridge length scales with $g \sim \sqrt{\lambda}$, then the bridge presence significantly affects the final result, and it would be interesting to reproduce this more general result from a string sigma-model minimal-area computation. If $\ell = \mathcal{O}(1)$, then $\mathcal{L} \to 0$, and the bridge has no effect at strong coupling, as expected.

# C  Minimal Areas Ending on Geodesics

This appendix followed from the following observation by Martin Kruczenski: *If we have some concatenation of geodesics on AdS, whose endpoints lie on a common circle, then the minimal surface which ends on those geodesics is nothing but the part of a spherical dome ending on the circle that is enclosed by the geodesics,*[19] see Figure 10a. That circle configuration can be mapped to the straight line, where the area is even simpler and given in Figure 10b.

It then becomes a straightforward exercise to compute this area. Of course, this problem is *not* the actual problem we want to solve, as here we are totally ignoring the sphere. Indeed, instead of obtaining the rich result (2.21) in the circle limit $z \to \bar{z}$ (or $\phi \to 0$), this simpler minimal area computation yields a simple constant, an integer multiple of $\pi$. As explained in the figure, for an $n$-point function we would simply need to consider the area of $n - 2$ world-sheet patches, each of which lies above its own geodesic circle over the straight line. Each of these thus gives

$$\int_{x_i}^{x_{i+1}} dx \int_b^{\infty} dz \frac{1}{z^2} = \pi, \qquad b = \sqrt{\left( \frac{x_i - x_{i+1}}{2} \right)^2 - \left( \frac{x_i + x_{i+1}}{2} - x \right)^2}, \qquad \text{(C.1)}$$

so that the area for $n$ such geodesics simply is $(n - 2)\pi$ in this sphere-free toy model. Note that each area piece is a pure number, independent of the geodesic end-points $x_i$; this is because of conformal invariance. Relatedly, note that this area bounded by geodesics is manifestly finite, without any need of subtractions, as anticipated in [2]. This is in contrast with other, more conventional minimal surface problems in $AdS/CFT$, where the surfaces go all the way to the boundary, thus picking up a divergent piece which one should re-normalize.

---

[19]This is in fact a general property of minimal surfaces: The condition for minimality (vanishing of the mean curvature) is a local condition. Hence cutting off arbitrary parts of any given minimal surface (in our case, a half-sphere in the Poincaré plane) again yields a minimal surface, with the boundary conditions given by the chosen cut contours.

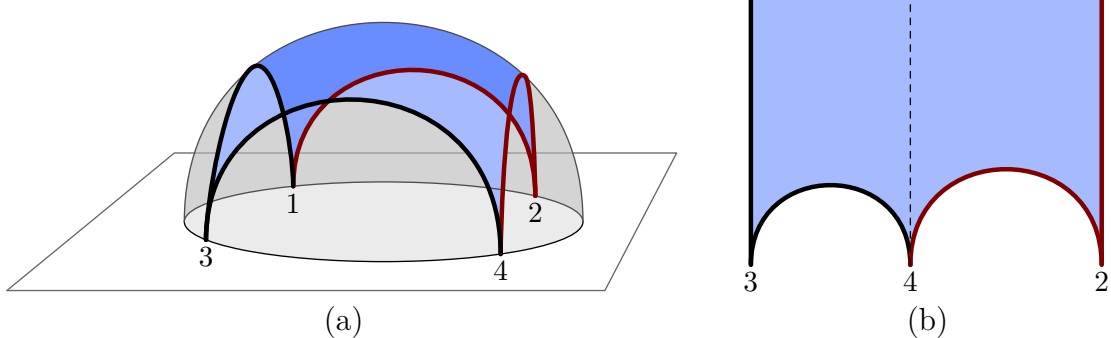

**Figure 10:** (a) Several geodesics ending on the same circle are conformally equivalent to (b) Geodesics ending on the same straight line. In the latter picture, we used conformal symmetry to put one of the operators at infinity. We see very clearly in this frame that the area becomes the sum of two pieces, separated by the dashed line. More general, had we started with $n$ points on a line, we would have ended with $n - 2$ such world-sheet patches. In the text, we show that the area of each patch is $\pi$. In the left figure removing the area below the geodesics amounts to removing the gray patches of the spherical dome, leaving only the blue cap.

Of course, our actual result for the area is not as simple, although it does simplify a little bit once we put all operators on a common line/circle: It becomes a simple function of $\varphi$:

$$\log \mathbb{O} \simeq \frac{\sqrt{\lambda}}{2\pi} \int\limits_{-\infty}^{\infty} \frac{d\theta}{2\pi} \, \varphi \cosh(\theta) \log \left[ 1 - \frac{\sinh^2\left(\frac{\varphi}{2}\right)}{\sinh^2\left(\frac{\varphi}{2}\cosh(\theta)\right)} \right]. \tag{C.2}$$

# D  Analytic Structure

After partial integration, the area (3.1) becomes (boundary terms vanish)

$$\log \mathbb{O} \simeq \frac{\sqrt{\lambda}}{2\pi} \int\limits_{-\infty}^{\infty} \frac{d\theta}{2\pi} \frac{\varphi^2 \sinh(\theta)^2 \sinh(\varphi \cosh \theta)(\cos \phi - \cosh \varphi)}{(\cosh \varphi - \cosh(\varphi \cosh \theta))(\cos \phi - \cosh(\varphi \cosh \theta))} \quad (\varphi < 0). \tag{D.1}$$

The integrand is now a meromorphic function of $\theta$, and most of the analytic structure of $\log \mathbb{O}$ can be inferred from the behavior of its poles and their residues. The poles of the integrand are the zeros of the two factors in the denominator, which are located at

$$\theta = \pm \operatorname{arccosh}\left(1 + \frac{2\pi i \mathbb{Z}_{\neq 0}}{\varphi}\right) + \pi i \mathbb{Z}, \qquad \theta = \pm \operatorname{arccosh}\left(\frac{i\phi + 2\pi i \mathbb{Z}}{\varphi}\right) + \pi i \mathbb{Z}. \tag{D.2}$$

At the points $\theta = \pi i \mathbb{Z}$, the numerator factor $\sinh^2(\theta)$ vanishes, canceling the zero of the denominator; we hence excluded those points above. In Euclidean kinematics, both $\varphi$ and $\phi$ are real ($\bar{z}$ is the complex conjugate of $z$). For all real values of $\varphi$ and $\phi$, all poles remain away from the real axis. Moreover, the locations of the poles (D.2) as well es their residues are invariant under $\phi \to \phi \pm 2\pi i$. Hence, the function $\log \mathbb{O}$ is a single-valued smooth function in Euclidean kinematics, as it should be.

In complexified kinematics, both $z$ and $\bar{z}$ are complex and independent of each other. Euclidean kinematics are located on the real section $\bar{z} = z^*$, where $z$ and $\bar{z}$ are complex conjugates. In contrast, $z$ and $\bar{z}$ are real and independent in Lorentzian kinematics.

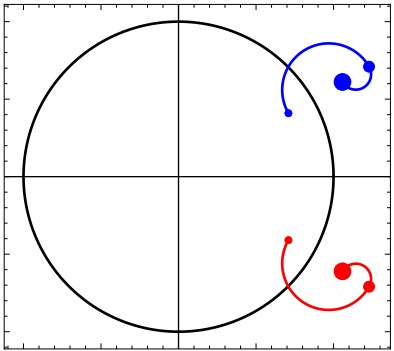

**Figure 11:** Inversion $\varphi \to -\varphi$ in the $z, \bar{z}$ plane: The figure shows the continuation path for $z$ (blue) and $\bar{z}$ (red). The continuation proceeds in four steps: (1) Blue segment from small to medium dot, (2) red segment from small to medium dot, (3) blue segment from medium to large dot, and (4) red segment from medium to large dot. This path of continuation avoids singular points where infinitely many residues accumulate at $\theta = \infty$. In this example, the three dots have coordinates $\phi = \pi/6$, $\varphi_1 = -1/5$, $\varphi_2 = 7/20$, $\varphi_3 = 1/5$.

## D.1 The Area is Even

In particular, the area should be invariant under inversions $z \to 1/\bar{z}$, $\bar{z} \to 1/z$, that is $\varphi \to -\varphi$. In contrast, the integrand of (D.1) behaves non-trivially as one passes from $\varphi < 0$ to $0 < \varphi$. Performing this continuation within the Euclidean section, almost all poles diverge to $\theta = \infty$ as $\varphi$ approaches $\varphi = 0$, which makes it difficult to see what happens during the continuation. We can circumvent this problem by deforming the continuation into complex kinematics. A convenient path of continuation is shown in Figure 11. Along such a path, all towers of poles stay at finite $\theta$, and two such towers cross the real line as shown in Figure 12. This deforms the contour of integration to the solid black line in the figure. Deforming it further to the dashed black line, we can relate the integral back to the original expression. To that end, first note that the integrand in (D.1) acquires a minus sign under a shift $\theta \to \theta \pm i\pi$. Hence the integrations along the horizontal dashed contours in Figure 12 equal *minus* the original integration along the real line. Moreover, the integrand is invariant under $\theta \to -\theta$; hence the integration along the vertical dashed contour from $-i\pi$ to $i\pi$ gives zero. Since the integrand in (D.1) is odd under $\varphi \to -\varphi$, one finds indeed that

$$\log \mathbb{O}(-\varphi) = \log \mathbb{O}(\varphi) \, , \tag{D.3}$$

as required by invariance under conformal inversions.

## D.2 Analytic Continuations

The complexified function $\log \mathbb{O}$ is locally holomorphic in $z$ and $\bar{z}$ independently. Globally, the function has branch points: For fixed $\bar{z}$ ($z$), there are branch points at $z = 0$ and $z = 1$ ($\bar{z} = 0$ and $\bar{z} = 1$). Because $\log \mathbb{O}$ is single-valued in Euclidean kinematics, the monodromies in $z$ and $\bar{z}$ are related:

$$\operatorname*{disc}_{z \circlearrowleft 0} \log \mathbb{O}(z, \bar{z}) + \operatorname*{disc}_{\bar{z} \circlearrowleft 0} \log \mathbb{O}(z, \bar{z}) = 0 \, , \qquad \operatorname*{disc}_{z \circlearrowleft 1} \log \mathbb{O}(z, \bar{z}) + \operatorname*{disc}_{\bar{z} \circlearrowleft 1} \log \mathbb{O}(z, \bar{z}) = 0 \, , \tag{D.4}$$

where e.g. $\operatorname{disc}_{z \circlearrowleft z_0} f(z)$ is the discontinuity that is picked up by $f(z)$ as $z$ follows a closed path encircling $z_0$ once counterclockwise, with all other variables held fixed.

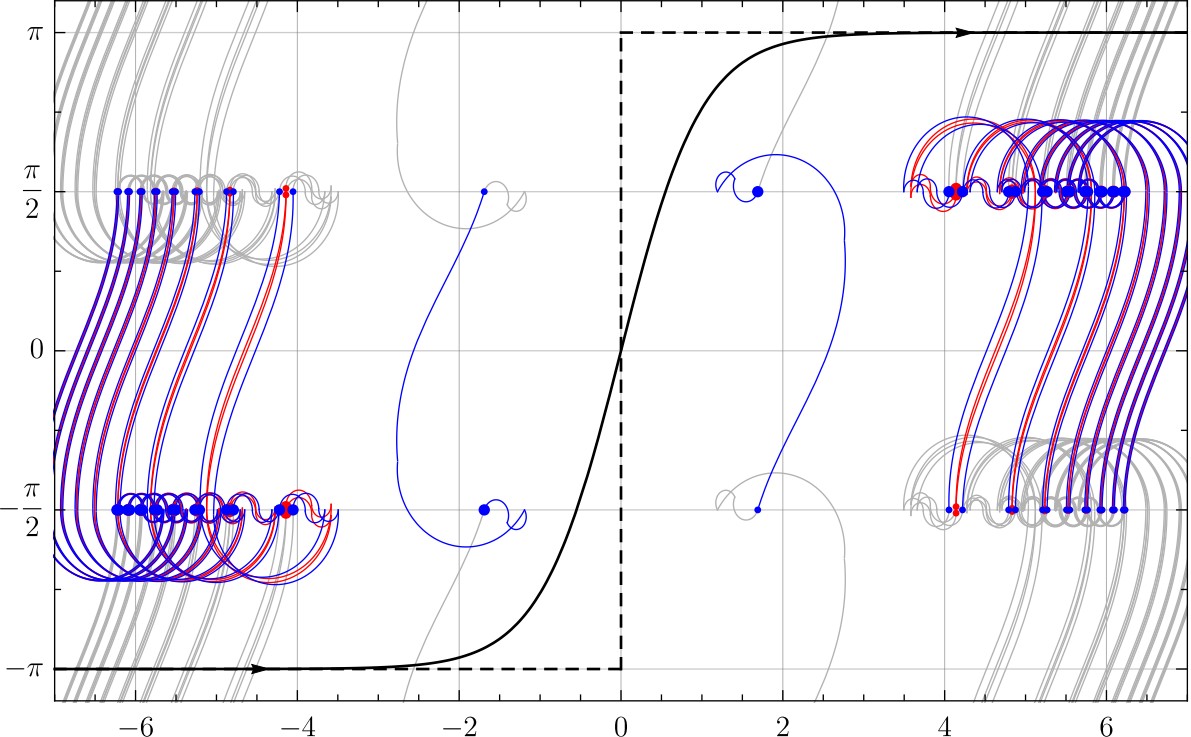

**Figure 12:** Inversion $\varphi \to -\varphi$ in the $\theta$ plane: During the continuation explained in Figure 11, the poles in the left/right half plane shift down/up along the contours shown (from small to large dots). This pushes the contour of integration to the black solid line. Deforming the contour further to the black dashed line, it can be easily related back to the original function, producing an overall minus sign.

**The Space of Analytic Continuations.** We want to explore analytic continuations of the area $\log \mathbb{O}$ as a complexified function of the two variables $z$ and $\bar{z}$. We will focus on continuations where $\bar{z}$ is held fixed, while $z$ follows some non-trivial cycles around $z = 0$ and/or $z = 1$. The extension to continuations that also involve cycles of $\bar{z}$ is not difficult, using relations such as (D.4). Without loss of generality, we will have our paths of analytic continuation start and end in the Lorentzian region $0 < z, \bar{z} < 1$. The space of analytic continuations of $\log \mathbb{O}$ is then a representation of the fundamental group of the sphere (compactified complex plane) with three marked points $z = \{0, 1, \infty\}$. The fundamental group is a free group with two generators. As our two generators, we choose $\mathcal{C}_0$ and $\mathcal{C}_1$ that wind $z$ counterclockwise around $z = 0$ and $z = 1$, respectively:

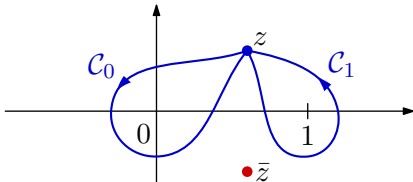

**The Branch Point at $z = 0$.** In the analytic continuation of $z$ around 0, the complete discontinuity comes from the map $z \mapsto \log z$ contained in the inversion of (1.2). In other words, $\varphi$ and $\phi$ provide uniformizing coordinates that resolve the branch point at $z = 0$ (and hence equally the branch point at $\bar{z} = 0$). In order to evaluate the continued function, we

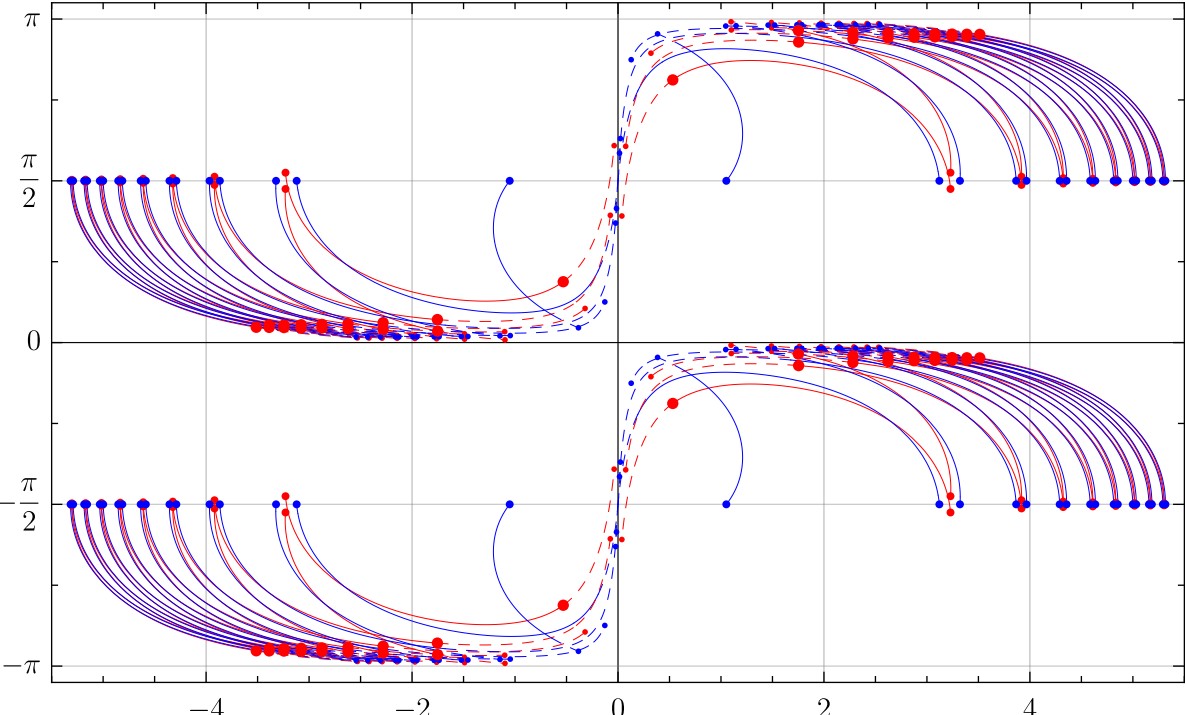

**Figure 13:** Continuation of $z$ around $z = 0$, with $\bar{z}$ held fixed. The figure shows the relevant poles of the integrand of $\log \mathbb{O}$ in the $\theta$ plane. The first sequence in (D.2) is shown in red, the second in blue. The path of continuation is $\varphi = 1/2 - i\alpha/2$, $\phi = \pi/5 + \alpha/2$; it starts and ends in the Euclidean section. Before the continuation, almost all poles lie near the lines $\mathrm{Im}(z) = \pm\pi/2$ (medium points). After one full rotation of $z$ ($\alpha = 2\pi$, large points), many poles have moved close to the real axis. After two more rotations of $z$ ($\alpha = 6\pi$, dashed lines, small points), some poles approach the imaginary axis. Going further, more and more poles accumulate near the imaginary axis. At no point does any pole cross the real line (contour of integration).

simply have to evaluate the integral (D.1) at appropriately shifted values of $\varphi$ and $\phi$. For example, a rotation of $z$ around the origin, with $\bar{z}$ held fixed, is realized by $\varphi \mapsto \varphi - i\pi$, $\phi \mapsto \phi + \pi$. As shown in Figure 13, no poles cross the real line during such continuations, and hence the $\theta$ contour of integration can be maintained without picking up any residues. Hence we find

$$\log \mathbb{O}(\varphi, \phi) \xrightarrow{(z \circlearrowleft 0)^n} \mathcal{C}_0^n \log \mathbb{O}(\varphi, \phi) = \log \mathbb{O}(\varphi - i\pi n, \phi + \pi n) \tag{D.5}$$

**The Branch Point at $z = 1$.** When we analytically continue in $z$ around $z = 1$, with $\bar{z}$ sufficiently far away from $\bar{z} = 1$, two poles cross the real axis, and hence the integral (D.1) picks up the residues of those poles, see Figure 14.[20] Starting (and ending) the continuation at $0 < \bar{z} < z < 1$ with $\varphi$ real and $\phi$ purely imaginary, the poles that cross the real axis are located at

$$\theta_{\pm} = \mp \mathrm{arccosh}\left(-\frac{\log z - \log \bar{z}}{\log z + \log \bar{z}}\right) = \mp \mathrm{arccosh}\left(\frac{i\phi}{\varphi}\right). \tag{D.6}$$

---

[20]Only these two poles cross as long as either $z\bar{z} < 0$ or $z\bar{z} > 0$ throughout the complete continuation. Otherwise, the end result (D.8) remains correct, but infinite towers of poles cross the real axis during the continuation, rendering the analysis slightly more complicated.

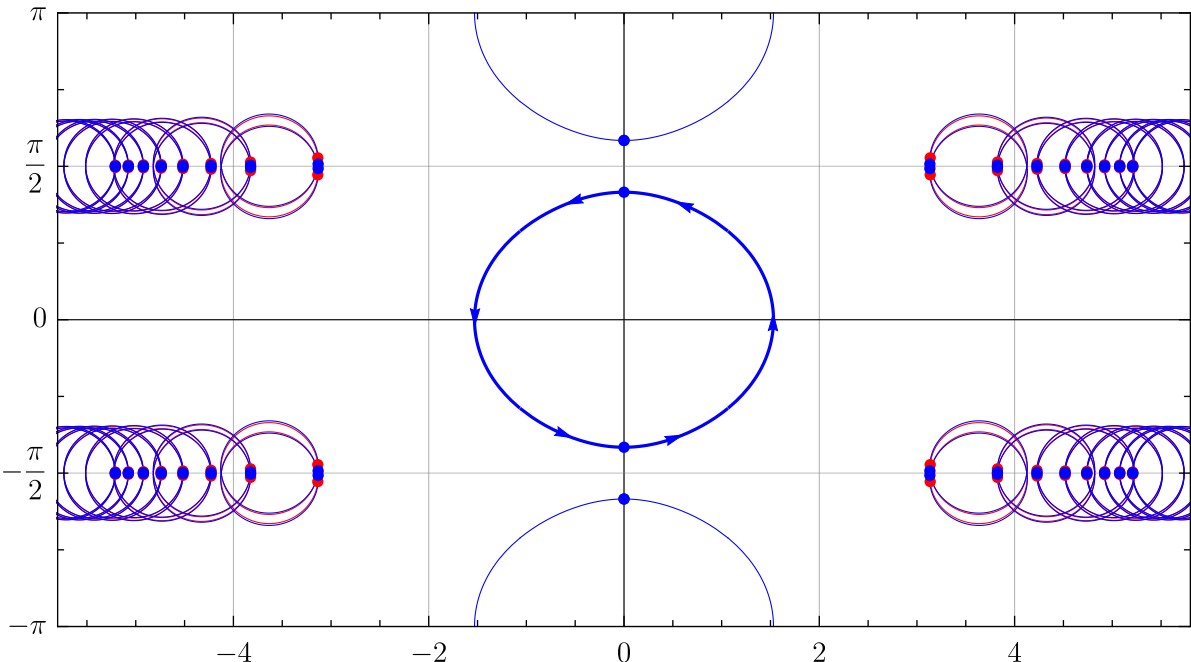

**Figure 14:** Continuation of $z$ around $z = 1$, with $\bar{z}$ held fixed. The figure shows the relevant poles of the integrand of $\log \mathbb{O}$ in the theta plane. The path of continuation in this example is $z = 1 - 1/3\, e^{i\alpha}$, $\bar{z} = 1/2$; it starts and ends in Lorentzian kinematics. Almost no poles come close to the real line during the continuation. The configurations before and after the continuation (at $\alpha = 0$ and $\alpha = 2\pi$) are identical, but two poles have crossed the real line and interchanged (shown in red and blue). Hence the function $\log \mathbb{O}$ picks up the residues of these poles.

They start on the imaginary axis, and rotate counterclockwise by 180°, ending up at their initial, but now interchanged, locations. At the beginning of the continuation, $\theta_+$ lies in the upper half plane, and $\theta_-$ lies in the lower half plane. Hence $\theta_+$ crosses the real axis from above, while $\theta_-$ crosses it from below. Before the continuation, the residues of the integrand in (D.1) at these poles are

$$\pm \frac{i}{2\pi} \sqrt{\log(z)\log(\bar{z})}\,. \tag{D.7}$$

But the continuation rotates $\log(z)$ around 0, hence the sign of the square root switches. Combining all signs and contour orientations, we therefore find

$$\log \mathbb{O} \xrightarrow{z \circlearrowleft 1} \mathcal{C}_1 \log \mathbb{O} = \log \mathbb{O} + \frac{\sqrt{\lambda}}{\pi} \sqrt{\log(z)\log(\bar{z})} \tag{D.8}$$

under the counterclockwise rotation of $z$ around $z = 1$ with $\bar{z}$ fixed. We find the same result when the start and end points of the continuation lie in the region $1 < z < \bar{z}$. At the end of the continuation, the poles (D.6) have interchanged, but their residues (D.7) have also swapped, so that again the residue with the plus sign is in the upper half plane, while the residue with the minus sign is in the lower half plane. When we now rotate back, i.e. apply $\mathcal{C}_1^{-1}$, the area $\log \mathbb{O}$ thus obtains the same discontinuity:

$$\log \mathbb{O} \xrightarrow{z \circlearrowleft 1} \mathcal{C}_1^{-1} \log \mathbb{O} = \log \mathbb{O} + \frac{\sqrt{\lambda}}{\pi} \sqrt{\log(z)\log(\bar{z})}\,. \tag{D.9}$$

This is consistent, as the discontinuity term is a square root that itself changes sign upon continuation:

$$\mathcal{C}_1^{-1} \frac{\sqrt{\lambda}}{\pi} \sqrt{\log(z)\log(\bar{z})} = \mathcal{C}_1 \frac{\sqrt{\lambda}}{\pi}\sqrt{\log(z)\log(\bar{z})} = -\frac{\sqrt{\lambda}}{\pi}\sqrt{\log(z)\log(\bar{z})}\,, \qquad \text{(D.10)}$$

such that consistently

$$\mathcal{C}_1^{-1}\mathcal{C}_1\log\mathbb{O} = \mathcal{C}_1^{-1}\log\mathbb{O} + \mathcal{C}_1^{-1}\frac{\sqrt{\lambda}}{\pi}\sqrt{\log(z)\log(\bar{z})} = \log\mathbb{O}\,. \qquad \text{(D.11)}$$

From the above discussion, it is clear what happens when $z$ winds around $z = 1$ any number of times: For each winding (in either direction), $\log\mathbb{O}$ picks up a term as in (D.8). In addition, an already present term of this type will change sign, thus canceling the new term. Hence we conclude that

$$\log\mathbb{O} \xrightarrow{(z\circlearrowleft 1)^n} \mathcal{C}_1^n \log\mathbb{O} = \log\mathbb{O} + \delta_{1,(n\bmod 2)}\frac{\sqrt{\lambda}}{\pi}\sqrt{\log(z)\log(\bar{z})}\,. \qquad \text{(D.12)}$$

**Combined Continuations.** We can now consider more complicated continuations that combine windings around $z = 0$ and $z = 1$. Suppose that we start in the Lorentzian section with $0 < (z, \bar{z}) < 1$, $\varphi$ real and $\phi$ imaginary, keep $\bar{z}$ fixed, and let $z$ undergo a sequence of windings around $z = 0$ and/or $z = 1$. This amounts to applying a sequence of continuations $\mathcal{C} = \ldots \mathcal{C}_1^{p_4}\mathcal{C}_0^{p_3}\mathcal{C}_1^{p_2}\mathcal{C}_0^{p_1}$ to $\log\mathbb{O}$. We saw above that $\mathcal{C}_0$ merely shifts $\varphi$ and $\phi$ in the formula (D.1) for $\log\mathbb{O}$, not affecting the contour of integration. In contrast, $\mathcal{C}_1$ picks up residues from two specific poles of the integrand, as shown in Figure 14. Now it turns out that when we first apply a number of continuations $\mathcal{C}_0^n$ around zero, and then continue around $z = 1$, a *different* pair of poles will cross the integration contour (see Figure 15). Namely, applying $\mathcal{C}_1\mathcal{C}_0^n$, the two poles that cross the integration contour are

$$\theta_{n,\pm} = \mp \operatorname{arccosh}\left(\frac{i\phi - 2\pi in}{\varphi}\right)\,. \qquad \text{(D.13)}$$

Again, $\theta_{n,+}$ crosses from the upper half plane, and $\theta_{n,-}$ crosses from the lower half plane. At the end of the continuation, the residues of the integrand in (D.1) at these poles are

$$\mp\frac{i}{2\pi}\sqrt{\left(\varphi + i\phi - 2\pi in\right)\left(\varphi - i\phi + 2\pi in\right)} = \mp\frac{i}{2\pi}\sqrt{\log(z)\left(\log(\bar{z}) + 2\pi in\right)}\,, \qquad \text{(D.14)}$$

where the logarithms are evaluated on the standard branch, that is $-i\pi < \operatorname{Im}(\log(\alpha)) \leq i\pi$. We therefore find

$$\mathcal{C}_0^n\log\mathbb{O} \xrightarrow{z\circlearrowleft 1} \mathcal{C}_1\mathcal{C}_0^n\log\mathbb{O} = \mathcal{C}_0^n\log\mathbb{O} + \frac{\sqrt{\lambda}}{\pi}\sqrt{\log(z)\left(\log(\bar{z}) + 2\pi in\right)}\,. \qquad \text{(D.15)}$$

The total effect of $n$ windings around $z = 0$ followed by one winding around $z = 1$ (all counterclockwise) hence is

$$\mathcal{C}_1\mathcal{C}_0^n\log\mathbb{O}(\varphi,\phi) = \log\mathbb{O}(\varphi - i\pi n, \phi + \pi n) + \frac{\sqrt{\lambda}}{\pi}\sqrt{\log(z)\left(\log(\bar{z}) + 2\pi in\right)}\,. \qquad \text{(D.16)}$$

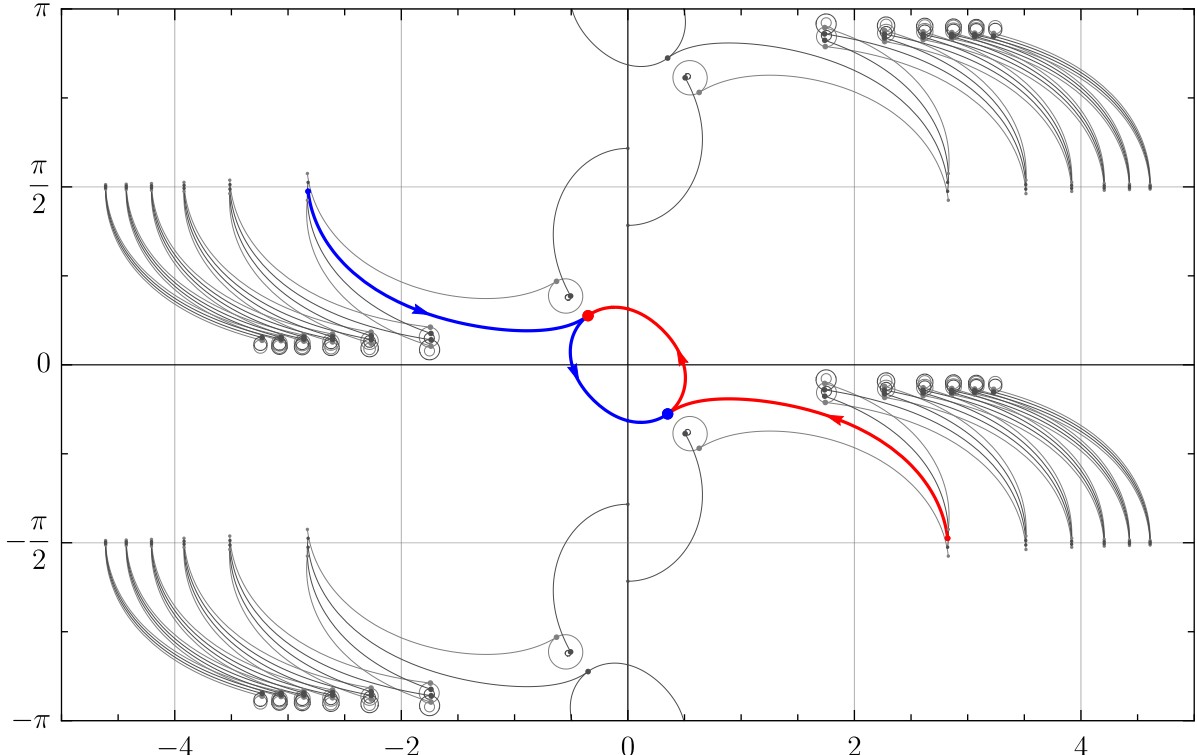

**Figure 15:** Motion of poles in the $\theta$ plane under the iterated continuation $\mathcal{C}_1\mathcal{C}_0$: Starting with $0 < \bar{z} < z < 1$, $z$ first follows counterclockwise circles, first around $z = 0$, then around $z = 1$. The poles that cross the integration contour are highlighted. All other poles are shown in gray. For a general continuation $\mathcal{C}_1\mathcal{C}_0^n$, the pair of poles that crosses the integration contour depends on the winding number $n$ of $z$ around $z = 0$, see (D.13).

As in the simple case (D.9), performing a reverse rotation around $z = 1$ yields the same result:

$$\mathcal{C}_1^{-1}\mathcal{C}_0^n \log \mathbb{O}(\varphi, \phi) = \log \mathbb{O}(\varphi - i\pi n, \phi + \pi n) + \frac{\sqrt{\lambda}}{\pi}\sqrt{\log(z)\big(\log(\bar{z}) + 2\pi in\big)}. \qquad \text{(D.17)}$$

In contrast, letting $z$ first wind once around $z = 1$ and then $n$ times around $z = 0$ yields

$$\mathcal{C}_0^n\mathcal{C}_1 \log \mathbb{O}(\varphi, \phi) = \log \mathbb{O}(\varphi - i\pi n, \phi + \pi n) + \frac{\sqrt{\lambda}}{\pi}\sqrt{\big(\log(z) + 2\pi in\big)\log(\bar{z})}. \qquad \text{(D.18)}$$

This clearly shows that the two generators $\mathcal{C}_0$ and $\mathcal{C}_1$ do not commute.

With these results, we can compute all analytic continuations in $z$: Any path of analytic continuation can be written as a product of factors $\mathcal{C}_1\mathcal{C}_0^n$ and $\mathcal{C}_1^{-1}\mathcal{C}_0^n$. The action of these factors on $\log \mathbb{O}$ is given above in (D.16) and (D.17), and the action on the extra terms

$$\Delta_{p,q} := \frac{\sqrt{\lambda}}{\pi}\sqrt{\big(\log(z) + 2\pi ip\big)\big(\log(\bar{z}) + 2\pi iq\big)} \qquad \text{(D.19)}$$

produced by preceding factors is simple:

$$\mathcal{C}_0^n\Delta_{p,q} = \Delta_{p+n,q}, \qquad \mathcal{C}_1^{\pm 1}\Delta_{p,q} = (1 - 2\delta_{p,0})\Delta_{p,q} = \begin{cases} -\Delta_{p,q} & p = 0, \\ +\Delta_{p,q} & p \neq 0. \end{cases} \qquad \text{(D.20)}$$

With these continuations, we can evaluate the area $\log \mathbb{O}$ in all Euclidean and Lorentzian sections in all kinematics. For example, with the shorthand

$$\log \mathbb{O}_n := \log \mathbb{O}(\varphi - i\pi n, \phi + \pi n)\,, \tag{D.21}$$

we find

$$
\begin{aligned}
\mathcal{C}_1^{\pm 1}\mathcal{C}_0^n\mathcal{C}_1^{\pm 1}\mathcal{C}_0^m \log \mathbb{O} &= \mathcal{C}_1^{\pm 1}\mathcal{C}_0^n\Big(\log \mathbb{O}_m + \Delta_{0,m}\Big) \\
&= \log \mathbb{O}_{m+n} + \Delta_{0,m+n} + \mathcal{C}_1^{\pm 1}\mathcal{C}_0^n\,\Delta_{0,m} \\
&= \log \mathbb{O}_{m+n} + \Delta_{0,m+n} + (1 - 2\delta_{n,0})\Delta_{n,m}\,. 
\end{aligned}
\tag{D.22}
$$

Combining these formulae, we arrive at the expressions (3.25) for the continuation of $\log \mathbb{O}$ from the Euclidean into any Lorentzian region.

# E   Weak Coupling Comparison

Table 2 summarizes the main similarities and differences between weak and strong coupling. See the main text for all details and precise pre-factors.

# F   Expansions of the Strongly-Coupled Octagon

This appendix is devoted to the derivation of the OPE limits of the octagon at strong coupling. We map this problem to the study of various limits of the integral:

$$\mathbb{I}(\varphi, \mu) \,=\, -\varphi \int_0^\infty d\theta \cosh\theta \log\!\Big(1 - e^\mu\, e^{-\varphi \cosh\theta}\Big) \,=\, -\varphi \int_1^\infty dt\, \frac{\log\left(1 - e^\mu\, e^{-\varphi t}\right)}{\sqrt{1 - 1/t^2}} \tag{F.1}$$

which serves as a building block for the octagon upon identifying the integrand (2.22) of this latter:

$$
\begin{aligned}
\log\left(1 + Y\right) &= \log \frac{\Big(1 - e^{-\log(z\bar z)/2}\, e^{\log(z\bar z)/2\,\cosh(\theta)}\Big)\Big(1 - e^{\log(z\bar z)/2}\, e^{\log(z\bar z)/2\,\cosh(\theta)}\Big)}{\Big(1 - e^{-\log(z/\bar z)/2}\, e^{\log(z\bar z)/2\,\cosh(\theta)}\Big)\Big(1 - e^{\log(z/\bar z)/2}\, e^{\log(z\bar z)/2\,\cosh(\theta)}\Big)} \\
&= \log \frac{\Big(1 - e^{\varphi}\, e^{-\varphi \cosh(\theta)}\Big)\Big(1 - e^{-\varphi}\, e^{-\varphi \cosh(\theta)}\Big)}{\Big(1 - e^{i\phi}\, e^{-\varphi \cosh(\theta)}\Big)\Big(1 - e^{-i\phi}\, e^{-\varphi \cosh(\theta)}\Big)}
\end{aligned}
\tag{F.2}
$$

can be decomposed into four pieces which give the following representation:

$$\log \mathbb{O} \,=\, -\frac{\sqrt{\lambda}}{2\pi^2}\left(\mathbb{I}(\varphi, \varphi) + \mathbb{I}(\varphi, -\varphi) - \mathbb{I}(\varphi, i\phi) - \mathbb{I}(\varphi, -i\phi)\right) \tag{F.3}$$

with cross ratios:

$$\varphi = -\frac{1}{2}\log\left(z\bar z\right) \quad \text{and} \quad i\phi = \frac{1}{2}\log\left(\frac{z}{\bar z}\right) \tag{F.4}$$

The closed form of the integral (F.1) is not known for arbitrary values of $\varphi$ and $\mu$, nevertheless we can obtain closed-form expressions in the limits where these parameters are very large or very small.

| Common Name | Kinematics | Strong Coupling $\mathbb{O} \sim e^{-\sqrt{\lambda}\mathbb{A}}$ | Perturbative Result |
|---|---|---|---|
| Euclidean neighbor OPE | $\bar{z}, z \to 0$ | $\mathbb{A} \sim \sqrt{\log(z\bar{z})}$ | $\mathbb{O} \simeq \sum \lambda^n \sum_{k=0}^{n} \log(z\bar{z})^k \sum_{r,p} z^r \bar{z}^p c_{k,n,r,p}$ |
| Euclidean diagonal OPE | $\bar{z}, z \to 1$ | $\mathbb{A} \sim \sqrt{(1-z)(1-\bar{z})}$ | $\mathbb{O} \simeq \sum \lambda^n \sum_{k=0}^{1} \log(y\bar{y})^k \sum_{r,p} y^r \bar{y}^p d_{k,n,r,p}$ |
| Double light-like neighbor OPE | $z \to 0$ and $\bar{z} \to \infty$ | $\mathbb{A} \sim \log(z/\bar{z})^2$ | $\log \mathbb{O} \simeq -\frac{(\log(-z)+\log(-1/\bar{z}))^2}{8\pi^2}\Gamma(\lambda) + \frac{1}{8}C(\lambda) + \frac{\lambda}{16\pi^2}\log(z\bar{z})^2$ |
| Double light-like diagonal OPE | $z \to 0, \bar{z} \to 1$ | $\mathbb{A} \sim \sqrt{\log(\frac{1}{z})}\sqrt{1-\bar{z}}$ | $\mathbb{O} \simeq \sum \lambda^n \sum_{k=0}^{n}\sum_{k'=0}^{1} \log(z)^k \log(\bar{y})^{k'} \sum_{r,p} z^r \bar{y}^p e_{k,k',n,r,p}$ |
| Single light-like neighbor OPE | $z \to 0$ | $\mathbb{A} \sim \sqrt{\log(\frac{1}{z})}(\mathrm{Li}_{\frac{3}{2}}(1) - \mathrm{Li}_{\frac{3}{2}}(\bar{z}))$ | $\mathbb{O} \simeq 1 - g^2 \frac{(1-\bar{z})(\log(z\bar{z})\log(1-\bar{z})+2\mathrm{Li}_2(\bar{z}))}{\bar{z}} + \cdots$ |
| Single light-like diagonal OPE | $\bar{z} \to 1$ | $\mathbb{A} \sim \sqrt{\log(\frac{1}{z})}\sqrt{1-\bar{z}}$ | $\mathbb{O} \simeq 1 + g^2 \bar{y}\left(\frac{-\pi^2}{3} - \log\bar{y}\log z + \log(1-z)\log z + 2\mathrm{Li}_2(z)\right) + \cdots$ |
| Diagonal equal neighboring length | $z\bar{z} \to 1$ | $\mathbb{A} \sim \log(\frac{z}{\bar{z}})(2\pi - \frac{1}{2i}\log(\frac{z}{\bar{z}}))$ | $\mathbb{O} =$ no particular simplification |
| Regge (Figure 9b) | $z, \bar{z} \to 0$ after $z \circlearrowleft 1$ | $\mathbb{A} \sim \sqrt{\log(z)\log(\bar{z})}$ | $\mathbb{O} \simeq 1 + g^2\, 2\pi i \frac{\log z - \log\bar{z}}{z - \bar{z}} + \cdots$ |
| Regge (Figure 9a) | $z, \bar{z} \to 1$ after $z \circlearrowleft 0$ | $\mathbb{A} \sim \sqrt{y} + \sqrt{\bar{y}} - \sqrt{y+\bar{y}}$ | $\mathbb{O} \simeq 1 + g^2\, 2\pi i\, y\bar{y}\, \frac{\log y - \log\bar{y}}{y - \bar{y}} + \cdots$ |
| Bulk point | $z \to \bar{z}$ after $z \circlearrowleft 0$ & $z \circlearrowleft 1$ | $\mathbb{A} \sim$ regular | $\mathbb{O} \simeq 1 + g^2\, 4\pi^2 \frac{(1-\bar{z})^2}{z - \bar{z}} + \cdots$ |

**Table 2:** Telegraphic summary of the differences and similarities between strong coupling and perturbation theory. We expect the strong coupling results to be most representative of the full finite coupling behavior. On the weak coupling column we use the short-hand notation: $y \equiv 1 - z$, $\bar{y} \equiv 1 - \bar{z}$ and $\bar{w} \equiv \frac{1}{\bar{z}}$

For instance the case $\mu = 0$ has been well studied, see [25], and the limit for large argument is known in terms of modified Bessel functions of the second kind:

$$\mathbb{I}(\varphi, 0) \overset{\varphi \to \infty}{=} \varphi \sum_{n=1}^{\infty} \frac{1}{n} \mathbf{K}_1(n\varphi) \tag{F.5}$$

This series is convergent and can be well approximated by the first few terms thanks to the exponential suppression $\mathbf{K}_1(n\varphi) \simeq \sqrt{\pi/2}\, e^{-n\varphi}/\sqrt{n\varphi}$ in the large limit $\varphi \to \infty$.

The small limit $\varphi \to 0$ is also known as:

$$\mathbb{I}(\varphi, 0) \overset{\varphi \to 0}{=} \frac{\pi^2}{6} - \frac{\pi}{2}\varphi + \varphi^2\left(\frac{1}{8} - \frac{\gamma_E}{4} - \frac{1}{4}\log\left(\frac{\varphi}{4\pi}\right)\right) + \sum_{k=1}^{\infty} \mathcal{C}_k\, \varphi^{2k+2} \tag{F.6}$$

but unlike the former case, this series has a finite radius of convergence as dictated by the coefficients:

$$\mathcal{C}_k = \frac{(-1)^{k+1}\zeta(2k+1)\Gamma(2k+1)}{2^{4k+2}\pi^{2k}\Gamma(k+1)\Gamma(k+2)} \text{ with } k \geq 1\,. \tag{F.7}$$

In the context of this paper the series in (F.5) and (F.6) give us access to OPE limits of our four-point function in the restricted kinematics $\phi = 0$ or $z = \bar{z}$. In order to address more generic OPE limits we need to find out how to incorporate the chemical potential $\mu$ in these series. The rest of this appendix is devoted to this task.

In Section F.1 we revisit the large $\varphi$ series (F.5), now including the chemical potential $\mu \neq 0$. This series allows us to obtain the Euclidean OPE limits in Table 1. In particular we obtain the full series of the OPE limits $z \to 0$ and/or $\bar{z} \to 0$. Furthermore, thanks to the competition between the four terms in (F.3), we also get access to the leading term of the OPE series $z \to 1$ or $\bar{z} \to 1$ (not both limits together).

The remaining sections concern the limit of the integrals necessary to obtain the double light-cone limit of the octagon $-i\phi \to \infty$ in the restricted kinematics $\varphi \to 0$. We start warming up in Section F.2.1 showing how to obtain the small $\varphi$ series in (F.6) without chemical potential $\mu = 0$. This we achieved by starting with the large $\varphi$ series in (F.5), consider the small $\varphi$ series expansion of the Bessel function $\mathbf{K}_1(n\varphi)$ and then finally exchange the order of sums to perform the re-summation over $n$ in (F.5). This latter step requires the use of a Zeta regularization and reproduces the result in (F.6)[21]. In the following sections we incorporate large and small chemical potential $\mu$ in the regime of small $\varphi$. In Section F.2.2 we consider a large chemical potential $\mu$ which gives us access to the leading term of the double light-cone limit of our four-point function under the identification $\mu = -i\phi \to \infty$. Finally in Section F.2.2 we consider the limit $\mu = \pm\varphi \to 0$ and obtain a series representations for the contributions of $\mathbb{I}(\varphi, \pm\varphi)$ showing how they modify the sub-leading term of the double light-cone limit.

In all these derivations for series in small $\varphi$ we perform dangerous steps such as exchanging order of sums and regularizing infinite sums. We do not fully justify them but have verified our results numerically in their corresponding regimes of validity.

## F.1 Series $\varphi \to \infty$ Including a Chemical Potential $\mu$

The first representation in (F.5) can be obtained by expanding the log in the integrand considering large $\varphi$ and then performing the integral for each term in the series. Each term

---

[21]Except for the linear term in $\varphi$, see discussion below (F.23)

evaluates to the modified Bessel function of the second kind $\mathbf{K}_1$. Following this recipe we can easily incorporate the chemical potential as

$$\mathbb{I}(\varphi,\mu) = \varphi \sum_{n=1}^{\infty} \frac{e^{n\mu}}{n} \int_1^{\infty} dt \, \frac{e^{-n\varphi t}}{\sqrt{1-1/t^2}} = \varphi \sum_{n=1}^{\infty} \frac{e^{n\mu}}{n} \mathbf{K}_1(n\varphi) \tag{F.8}$$

This representation is suitable for large $\varphi$ since $\mathbf{K}_1(n\varphi) \simeq \frac{e^{-n\varphi}}{\sqrt{\pi\varphi}} + \cdots$.

In order to write an explicit series in $\frac{1}{\varphi}$ we first introduce the large $\varphi$ expansion of the Bessel function:

$$\mathbf{K}_1(n\varphi) = \sqrt{\varphi}\sqrt{\frac{\pi}{2}}\frac{e^{-n\varphi}}{\sqrt{n}} \sum_{k=1}^{\infty} c_k \left(\frac{1}{2n\varphi}\right)^{k-1} \quad \text{with} \quad c_k = (-1)^k \frac{(2k-1)!!(2k-5)!!}{2^{2k-2}(k-1)!} \tag{F.9}$$

Plugging this latter series into (F.8) and exchanging the sums we have:

$$\mathbb{I}(\varphi,\mu) = \sqrt{\varphi}\sqrt{\frac{\pi}{2}} \sum_{n=1}^{\infty} \frac{e^{\mu}}{n} \frac{e^{-n\varphi}}{\sqrt{n}} \sum_{k=1}^{\infty} c_k \left(\frac{1}{2n\varphi}\right)^{k-1}$$

$$\mathbb{I}(\varphi,\mu) = \sqrt{\varphi}\sqrt{\frac{\pi}{2}} \sum_{k=1}^{\infty} \frac{c_k}{(2\varphi)^{k-1}} \, \mathrm{Li}_{k+\frac{1}{2}}(e^{\mu-\varphi}) \tag{F.10}$$

Using this latter representation we find the $\varphi \to \infty$ or $z\bar{z} \to 0$ limit of our correlator as:

$$\log\mathbb{O} = -\frac{\sqrt{\lambda}}{4\pi^{3/2}}\sqrt{-\log z\bar{z}} \, \cdot$$
$$\cdot \sum_{k=1}^{\infty} \frac{c_k}{(-\log z\bar{z})^{k-1}} \left( \mathrm{Li}_{k+\frac{1}{2}}(1) + \mathrm{Li}_{k+\frac{1}{2}}(z\bar{z}) - \mathrm{Li}_{k+\frac{1}{2}}(z) - \mathrm{Li}_{k+\frac{1}{2}}(\bar{z}) \right) \tag{F.11}$$

This is a good representation for the limits $z \to 0$ or $\bar{z} \to 0$ or both $z,\bar{z} \to 0$. In particular its leading terms reproduces the corresponding Euclidean OPE limits in Table 1.

Furthermore, thanks to the competition between the four functions in (F.11), the first term of the series ($k=1$) provides the correct leading term of the limits $z \to 1$ or $\bar{z} \to 1$ or both $z,\bar{z} \to 1$. For instance for $z \to 1$ the leading term reproduces the result in Table 1:

$$\log\mathbb{O} \stackrel{z\to 1}{=} -\frac{\sqrt{\lambda}}{2\pi}\sqrt{1-z}\sqrt{-\log\bar{z}} \quad \text{and} \quad \log\mathbb{O} \stackrel{z,\bar{z}\to 1}{=} -\frac{\sqrt{\lambda}}{2\pi}\sqrt{1-z}\sqrt{1-\bar{z}} \tag{F.12}$$

However the rest of the series does not provide a good expansion in this limit. This can be noticed when trying to compute the first sub-leading term in the limit $z \to 1$, which requires a re-summation of all the series for finite $\bar{z}$. While for $z,\bar{z} \to 1$ the series is no longer convergent.

$$\log\mathbb{O} \stackrel{z\to 1}{=} -\frac{\sqrt{\lambda}}{2\pi}\sqrt{1-z}\sqrt{-\log\bar{z}} - \frac{\sqrt{\lambda}}{4\pi^{3/2}}(1-z)\sqrt{-\log\bar{z}} \, \cdot$$
$$\cdot \sum_{n=1}^{\infty} \frac{c_n}{(\log\bar{z})^{n-1}} \left( \mathrm{Li}_{n-\frac{1}{2}}(1) - \mathrm{Li}_{n-\frac{1}{2}}(\bar{z}) \right) + \mathcal{O}(1-z)^{\frac{3}{2}} \tag{F.13}$$

## F.2 The Double Light-Cone Limit $-i\phi \to \infty$ with $\varphi \to 0$

In this appendix, we consider the light-cone limit $z \to 0^-$, $\bar{z} \to -\infty$ in a restricted kinematics with small $\varphi$ i.e. $z\bar{z} \simeq 1$. To address this limit, we find it more convenient to make the arguments of the logarithms explicitly positive:

$$\log\left(1+Y\right) = \log \frac{\left(1 - e^{-\frac{\log(z\bar{z})}{2}} e^{\frac{\log(z\bar{z})}{2} \cosh(\theta)}\right)\left(1 - e^{\frac{\log(z\bar{z})}{2}} e^{\frac{\log(z\bar{z})}{2} \cosh(\theta)}\right)}{\left(1 + e^{\frac{1}{2}\left(\log(-z)+\log(-\frac{1}{\bar{z}})\right)} e^{\frac{\log(z\bar{z})}{2} \cosh(\theta)}\right)\left(1 + e^{-\frac{1}{2}\left(\log(-z)+\log(-\frac{1}{\bar{z}})\right)} e^{\frac{\log(z\bar{z})}{2} \cosh(\theta)}\right)} \tag{F.14}$$

and in order to avoid dealing with $i\pi$ shifts on the chemical potential we introduce a slightly modified integral:

$$\mathbb{I}_+(\varphi, \mu) = -\varphi \int_0^\infty d\theta \cosh\theta \log\left(1 + e^\mu\, e^{-\varphi\cosh\theta}\right) = -\varphi \int_1^\infty dt\, \frac{\log(1 + e^\mu\, e^{-\varphi\, t})}{\sqrt{1 - 1/t^2}} \tag{F.15}$$

Using this new building block we rewrite the octagon as:

$$\log \mathbb{O} = \frac{\sqrt{\lambda}}{2\pi^2}\left(\mathbb{I}_+(\varphi, \mu_{\text{light}}) + \mathbb{I}_+(\varphi, -\mu_{\text{light}}) - \mathbb{I}(\varphi, \varphi) - \mathbb{I}(\varphi, -\varphi)\right) \tag{F.16}$$

and the double-light cone limit is obtained with $\mu_{\text{light}} \equiv -\frac{1}{2}\left(\log(-z) + \log(-\frac{1}{\bar{z}})\right) \to +\infty$.

The first integral $\mathbb{I}_+(\varphi, \mu_{\text{light}})$ gives the leading contribution and is obtained in section F.2.2 in a $\mu \to \infty$ and $\varphi \to 0$ expansion. The second integral $\mathbb{I}_+(\varphi, -\mu_{\text{light}})$ is exponentially suppressed $\mathcal{O}(e^{-\mu})$ and can be neglected. The tail $-\mathbb{I}(\varphi, \varphi) - \mathbb{I}(\varphi, -\varphi)$ corrects the sub-leading term in the double light-cone limit and is obtained in section F.2.3 in a $\varphi \to 0$ expansion. Finally we put these results together in section F.2.4 reproducing (3.12).

Before addressing the relevant integrals for this light-cone limit, we start by deriving the small $\varphi \to 0$ expansion in (F.6) with $\mu = 0$. This exercise teaches us the steps such as sum exchanges and regularization necessary to find the series expansions of the integrals in (F.16).

### F.2.1 Series $\varphi \to 0$ without a Chemical Potential

Here we derive the series expansion in small $\varphi$ for $\mu = 0$ given in (F.6). For this we start with the representation (F.5) and consider the small $\varphi$ expansion of the Bessel function:

$$\frac{\varphi \mathbf{K}_1(n\varphi)}{n} \stackrel{\varphi \to 0}{=} \frac{1}{n^2} + \sum_{k=0}^\infty \varphi^{2k+2}\left(c_k\, n^{2k}\log n + \left(c_k \log\left(\frac{\varphi}{2}\right) - d_k\right)n^{2k}\right) \tag{F.17}$$

with:

$$c_k = \frac{1}{2^{2k+1}}\frac{1}{\Gamma(k+1)\Gamma(k+2)} \qquad \text{and} \qquad d_k = \frac{1}{2^{2k+2}}\frac{\psi(k+1) + \psi(k+2)}{\Gamma(k+1)\Gamma(k+2)}\,. \tag{F.18}$$

Combining this latter representation (F.17) for $\mathbf{K}_1$ with the series representation in (F.5) we want to derive the series representation in (F.6):

$$\lim_{\varphi \to 0} \mathbb{I}(\varphi, 0) = \sum_{n=1}^\infty \left(\lim_{\varphi \to 0} \frac{\varphi \mathbf{K}_1(n\varphi)}{n}\right)$$

$$= \sum_{n=1}^{\infty} \left( \sum_{k=0}^{\infty} \cdots \text{ in (F.17)} \right) \tag{F.19}$$

This requires exchanging the sums in $n$ and $k$. The leading term $\varphi^0$ comes from the sum:

$$\sum_{n=1}^{\infty} \frac{1}{n^2} = \zeta(2) = \frac{\pi^2}{6} \tag{F.20}$$

The coefficients of the subleading terms in $\varphi$ seem to diverge but performing a zeta-regularization we obtain the finite contributions:

$$\sum_{n=1}^{\infty} n^{2k} \log n = -\zeta'(-2k) = \begin{cases} \frac{\log 2\pi}{2} & \text{if } k = 0 \\ \frac{(-1)^{k+1}\zeta(2k+1)\Gamma(2k+1)}{2(2\pi)^{2k}} & \text{if } k \in \mathbb{Z}^+ \end{cases} \tag{F.21}$$

$$\sum_{n=1}^{\infty} n^{2k} = \zeta(-2k) = \begin{cases} -\frac{1}{2} & \text{if } k = 0 \\ 0 & \text{if } k \in \mathbb{Z}^+ \end{cases} \tag{F.22}$$

Combining these regularized sums we obtain the series expansion in (F.6) as:

$$\sum_{n=1}^{\infty} \frac{\varphi \mathbf{K}_1(n\varphi)}{n} \overset{\varphi \to 0}{=} \frac{\pi^2}{6} + \sum_{k=0}^{\infty} \varphi^{2k+2} \left( c_k \left( -\zeta'(-2k) \right) + \left( c_k \log\left( \frac{\varphi}{2} \right) - d_k \right) \zeta(-2k) \right) \tag{F.23}$$

$$= \frac{\pi^2}{6} + \varphi^2 \left( c_0 \frac{\log 2\pi}{2} + \left( c_0 \log\left( \frac{\varphi}{2} \right) - d_0 \right)\left( -\frac{1}{2} \right) \right) + \sum_{k=1}^{\infty} \left( -\zeta'(-2k) c_k \right) \varphi^{2k+2}$$

$$+ \sum_{k=1}^{\infty} \varphi^{2k+2} \left( c_k \log\left( \frac{\varphi}{2} \right) - d_k \right) \zeta(-2k) \tag{F.24}$$

$$= \frac{\pi^2}{6} - \frac{\pi}{2}\varphi + \varphi^2 \left( \frac{1}{8} - \frac{\gamma_E}{4} - \frac{1}{4} \log\left( \frac{\varphi}{4\pi} \right) \right) + \sum_{k=1}^{\infty} \mathcal{C}_k \varphi^{2k+2} \tag{F.25}$$

where we have used $\mathcal{C}_k = -\zeta'(-2k) c_k$ and $\zeta(-2k) = 0$ for positive integer $k$.

**Remark.** The red term in (F.25) was added by hand. It does not come naturally out of this illegal derivation with the dangerous summation manipulations performed above. All other terms come out perfectly and beautifully agree with the expansion quoted in [25]. In this case we could easily fix the linear term since it is the sub-leading term. Together with the logarithmic term, it is a kind of anomaly in the sense that it is the only term which does not respect the $\varphi \to -\varphi$ symmetry of the result. Since it is the subleading term in the small $\varphi$ expansion, it is trivial to restore it by an independent small $\varphi$ analysis. Now, in what follows we will carry on similar derivations for the case of interest with chemical potentials. We will again be missing the analogue of these linear red terms which we should add back at the end. However, we are in better shape here, since we know that the full result is actually $\varphi \to -\varphi$ invariant, as this is an exact symmetry of the octagon. As such, these linear terms must cancel when adding up all four free energies. Indeed we checked that they do. With this hindsight, we will thus ignore them completely in the discussion that follows. Let us also stress again that in the end all final expansions are carefully checked numerically anyway.

### F.2.2 Including a Large Chemical Potential $\mu \to \infty$

Now we study the slightly modified integral $\mathbb{I}_+(\varphi, \mu)$ of (F.15) in the combined limits $\mu \to \infty$ and $\varphi \to 0$. For this we start with the series expansion:

$$\mathbb{I}_+(\varphi, \mu) = \sum_{n=1}^{\infty} \frac{\varphi \mathbf{K}_1(n\varphi)\,(-e^\mu)^n}{n} \tag{F.26}$$

After taking the small $\varphi$ limit of the summand in (F.8) we obtain:

$$\frac{\varphi \mathbf{K}_1(n\varphi)\,(-e^\mu)^n}{n} \overset{\varphi \to 0}{=} \\ \frac{(-e^\mu)^n}{n^2} + \sum_{k=0}^{\infty} \varphi^{2k+2} \left( c_k\,(-e^\mu)^n n^{2k} \log n + \left( c_k \log\left(\frac{\varphi}{2}\right) - d_k \right)(-e^\mu)^n\, n^{2k} \right) \tag{F.27}$$

The sums on $n$ can be performed using a generalization of the Zeta-regularization which now gives polylogarithms $\mathrm{Li}_k(x)$ and their derivatives[22] $\mathrm{Di}_k(x) \equiv -\partial_m \mathrm{Li}_m(x)\big|_{m \to k}$,

$$\sum_{n=1}^{\infty} \frac{\varphi \mathbf{K}_1(n\varphi)\,(-e^\mu)^n}{n} \overset{\varphi \to 0}{=} \mathrm{Li}_2(-e^\mu) + \varphi^2 \left( \mathrm{Di}_0(-e^\mu)\, c_0 + \mathrm{Li}_0(-e^\mu) \left( c_0 \log\left(\frac{\varphi}{2}\right) - d_0 \right) \right) \\ + \sum_{k=1}^{\infty} \varphi^{2k+2} \left( \mathrm{Di}_{-2k}(-e^\mu)\, c_k + \mathrm{Li}_{-2k}(-e^\mu) \left( c_k \log\left(\frac{\varphi}{2}\right) - d_k \right) \right) \tag{F.28}$$

In the large $\mu$ limit these regularized sums behave as:

$$\sum_{n=1}^{\infty} \frac{(-e^\mu)^n}{n^2} = \mathrm{Li}_2(-e^\mu) \overset{\mu \to \infty}{=} -\frac{\mu^2}{2} - \frac{\pi^2}{6} + \mathcal{O}(e^{-\mu}) \tag{F.29}$$

$$\sum_{n=1}^{\infty} (-e^\mu)^n\, n^{2k} \log n = \mathrm{Di}_{-2k}(-e^\mu) \overset{\mu \to \infty}{=} \begin{cases} \log \mu + \gamma_E + \mathcal{O}(\frac{1}{\mu^2}) & \text{if } k = 0 \\ -\frac{\Gamma(2k)}{\mu^{2k}} + \mathcal{O}(\frac{1}{\mu^{2k+2}}) & \text{if } k \in \mathbb{Z}^+ \end{cases} \tag{F.30}$$

$$\sum_{n=1}^{\infty} (-e^\mu)^n\, n^{2k} = \mathrm{Li}_{-2k}(-e^\mu) = \partial_\mu^{2k} \frac{-e^\mu}{1 + e^\mu} \overset{\mu \to \infty}{=} \begin{cases} -1 + e^{-\mu} + \mathcal{O}(e^{-2\mu}) & \text{if } k = 0 \\ e^{-\mu} + \mathcal{O}(e^{-2\mu}) & \text{if } k \in \mathbb{Z}^+ \end{cases} \tag{F.31}$$

The derivative of the polylogarithm $\mathrm{Di}_0$ contains power law corrections given by:

$$\lim_{\mu \to \infty} \mathrm{Di}_0(-e^\mu) = \log \mu + \gamma_E - \sum_{m=1}^{\infty} \frac{\left( 2 - \frac{1}{2^{2m-2}} \right) \Gamma(2m)\, \zeta(2m)}{\mu^{2m}} + \cdots \tag{F.32}$$

where the ellipsis stands for terms further suppressed exponentially $e^{-\mu}$. This function can be used as a seed to obtain the derivatives with negative indices from the recursion relation: $\partial_\mu^{2k} \mathrm{Di}_0(-e^\mu) = \mathrm{Di}_{-2k}(-e^\mu)$. This relation makes obvious the logarithmic divergence $\log \mu$ disappear and we only have power-suppressed corrections from these terms:

$$\lim_{\mu \to \infty} \mathrm{Di}_{-2k}(-e^\mu) = \frac{-\Gamma(2k)}{\mu^{2k}} - \sum_{m=1}^{\infty} \frac{\left( 2 - \frac{1}{2^{2m-2}} \right) \Gamma(2m + 2k)\, \zeta(2m)}{\mu^{2m+2k}} + \cdots \qquad \text{with } k \in \mathbb{Z}^+ \tag{F.33}$$

---

[22]Notice the derivative is over the index of the Polylogarithm and not on its argument.

Finally, neglecting exponentially suppressed terms $\mathcal{O}(e^{-\mu})$ we have the series:

$$\lim_{\mu\to\infty} \mathbb{I}_+(\varphi,\mu) \simeq -\frac{\mu^2}{2} - \frac{\pi^2}{6} + \varphi^2 \left( \mathrm{Di}_0(-e^\mu)\, c_0 + (-1)\left(c_0 \log\left(\frac{\varphi}{2}\right) - d_0\right)\right)$$
$$+ \sum_{k=1}^\infty \varphi^{2k+2}\mathrm{Di}_{-2k}(-e^\mu)\, c_k \tag{F.34}$$

More simply, neglecting $\mathcal{O}(1/\mu^2)$ corrections, we obtain:

$$\lim_{\mu\to\infty} \mathbb{I}_+(\varphi,\mu) \simeq -\frac{\mu^2}{2} - \frac{\pi^2}{6} + \varphi^2 \left( \frac{\log\mu}{2} + \frac{1}{4} - \frac{\log(\varphi/2)}{2}\right) \tag{F.35}$$

We thank Nikolay Gromov for providing us with a clean alternative derivation of these large chemical potential results which was extremely useful in cross-checking these manipulations.

### F.2.3 Series $\varphi \to 0$ with a Chemical Potential $\mu = \pm\varphi$

In this section we consider the contribution of $\mathbb{I}(\varphi,\varphi) + \mathbb{I}(\varphi,-\varphi)$ as a series expansion in small $\varphi$. For this we introduce the chemical potentials as a series expansion:

$$e^{n\varphi} + e^{-n\varphi} = \sum_{l=0}^\infty f_l \varphi^{2l} \quad \text{with } f_l = 2\Gamma(2l). \tag{F.36}$$

inside:

$$\mathbb{I}(\varphi,\varphi) + \mathbb{I}(\varphi,-\varphi) = \sum_{n=1}^\infty \frac{\varphi(e^{n\varphi} + e^{-n\varphi})\mathbf{K}_1(n\varphi)}{n} \overset{\varphi\to 0}{=} \cdot \sum_{n=1}^\infty \left( \frac{\sum_{l=0}^\infty f_l \varphi^{2l}}{n^2} + S_n^{\text{extra}}\right) \tag{F.37}$$

The leading term can be obtained after regularizing the sum over $n$ as:

$$\sum_{n=1}^\infty \sum_{l=0}^\infty f_l \varphi^{2l} n^{2l-2} = \sum_{l=0}^\infty \varphi^{2l}\, f_l\, \zeta(2-2l) = \frac{\pi^2}{3} - \frac{\varphi^2}{2} \tag{F.38}$$

The subleading terms come from $S_n^{extra}$ which, after rearranging the sums on $k$ and $l$, is given by:

$$S_n^{\text{extra}} = \sum_{k=0}^\infty \varphi^{2k+2}\left(\sum_{l=0}^\infty f_l\, n^{2l}\varphi^{2l}\right)\left(c_k\, n^{2k}\log n + \left(c_k \log\left(\frac{\varphi}{2}\right) - d_k\right)n^{2k}\right) \tag{F.39}$$
$$= \sum_{k=0}^\infty \sum_{l=0}^\infty \varphi^{2k+2l+2}\left(n^{2k+2l}\log n\, f_l\, c_k + n^{2k+2l}\left(f_l\, c_k \log\left(\frac{\varphi}{2}\right) - f_l\, d_k\right)\right)$$
$$= \sum_{M=0}^\infty \varphi^{2M+2}\left(n^{2M}\log n \sum_{k=0}^M f_{M-k}\, c_k + n^{2M}\sum_{k=0}^M\left(f_{M-k}\, c_k \log\left(\frac{\varphi}{2}\right) - f_{M-k}\, d_k\right)\right)$$

and performing the sum over $n$ using zeta-regularization we obtain:

$$\sum_{n=1}^\infty S_n^{extra} = \sum_{M=0}^\infty \varphi^{2M+2}\left( -\zeta'(-2M)\sum_{k=0}^M f_{M-k}\, c_k + \zeta(-2M)\sum_{k=0}^M f_{M-k}\left(c_k \log\left(\frac{\varphi}{2}\right) - d_k\right)\right)$$
$$= \varphi^2 f_0 \left(\frac{\log 2\pi}{2}c_0 - \frac{1}{2}\left(c_0 \log\left(\frac{\varphi}{2}\right) - d_0\right)\right) + \sum_{M=1}^\infty \varphi^{2k+2}\left(-\zeta'(-2M)\sum_{k=0}^M f_{M-k}\, c_k\right)$$
$$\tag{F.40}$$

Adding up (F.38) and (F.40) we get

$$\mathbb{I}(\varphi, \varphi) + \mathbb{I}(\varphi, -\varphi) = \sum_{n=1}^{\infty} \frac{\varphi(e^{n\varphi} + e^{-n\varphi})\mathbf{K}_1(n\varphi)}{n}$$

$$\stackrel{\varphi \to 0}{=} \frac{\pi^2}{3} + \varphi^2 \left( -\frac{1}{2} + \frac{\log 2\pi}{2} f_0 c_0 + \frac{1}{2} f_0 d_0 - \frac{1}{2} f_0 c_0 \log\left(\frac{\varphi}{2}\right) \right)$$

$$+ \sum_{M=1}^{\infty} \varphi^{2k+2} \left( -\zeta'(-2M) \sum_{k=0}^{M} f_{M-k} c_k \right) \tag{F.41}$$

Finally neglecting $\mathcal{O}(\varphi^4)$ terms we obtain:

$$\mathbb{I}(\varphi, \varphi) + \mathbb{I}(\varphi, -\varphi) \stackrel{\varphi \to 0}{=} \frac{\pi^2}{3} + \varphi^2 \left( -\frac{1}{2} \log \frac{\varphi}{4\pi} - \frac{1}{4} - \frac{\gamma_E}{2} \right) \tag{F.42}$$

### F.2.4   The Double Light-Cone Limit

The double light-cone limit corresponds to the limit $i\phi \to \infty$. Considering also the limit $\varphi \to 0$, we can provide an analytic expression for the octagon (F.16) by combining formulas (F.42) and (F.35) with $\mu \to \mu_{\text{light}} \equiv -\frac{1}{2}\left( \log(-z) + \log(-\frac{1}{\bar{z}}) \right)$:

$$\log \mathbb{O} = \frac{\sqrt{\lambda}}{2\pi^2}\left( \mathbb{I}_+(\varphi, \mu_{\text{light}}) + \mathbb{I}_+(\varphi, -\mu_{\text{light}}) - \mathbb{I}(\varphi, \varphi) - \mathbb{I}(\varphi, -\varphi) \right)$$

$$\simeq \frac{\sqrt{\lambda}}{2\pi^2}\left( -\frac{\mu_{\text{light}}^2}{2} - \frac{\pi^2}{6} + \varphi^2\left( \frac{\log \mu_{\text{light}}}{2} + \frac{1}{4} - \frac{\log(\varphi/2)}{2} \right) \right.$$

$$\left. - \left( \frac{\pi^2}{3} + \varphi^2\left( -\frac{1}{2} \log \frac{\varphi}{4\pi} - \frac{1}{4} - \frac{\gamma_E}{2} \right) \right) \right)$$

$$= \frac{\sqrt{\lambda}}{2\pi^2}\left( -\frac{\mu_{\text{light}}^2}{2} - \frac{\pi^2}{2} + \varphi^2\left( \frac{\log \mu_{\text{light}}}{2} + \frac{1 + \log 2 + \gamma_E - \log 4\pi}{2} \right) \right)$$

$$= -\frac{(-\log(-z) - \log(-1/\bar{z}))^2}{8\pi^2}\left( \frac{\sqrt{\lambda}}{2} \right) + \frac{1}{8}\left( -2\sqrt{\lambda} \right) \tag{F.43}$$

$$+ \frac{\sqrt{\lambda}}{16\pi^2}(\log z\bar{z})^2 \left( \log\left( -\log(-z) - \log(-1/\bar{z}) \right) + 1 + \gamma_E - \log 4\pi \right).$$

We have neglected terms suppressed as $\mathcal{O}(1/\mu_{\text{light}}^2, e^{-\mu_{\text{light}}}, \varphi^4)$ in order to compare with (3.12) and (3.13) in the main text. The power-like corrections on $\mathcal{O}(1/\mu_{\text{light}}^2)$ and $\mathcal{O}(\varphi^2)$ can be recovered from the series in (F.34) and (F.41).

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
