# Peer review of "Octagons II: Strong Coupling"

_SciPost Physics_

## Round 1 · Referee Report · Anonymous (Referee 1) · 2025-12-9

Strengths

intersting topic in 2019

Weaknesses

outdated topic in 2025

Report

I cannot recommend this manuscript for publication in its current form. It has to be updated/reworked in light of the above comments to be reconsidered. (see Requested changes)

Requested changes

Report on the manuscript "Octagons II" by Bargheer et al.

The topic of the octagon form factor is interesting, and the work performed is good; however, the presentation is obsolete. I find myself in a perplexing situation since it appears that the manuscript has been shelved for five years, so it is severely `stale'. I am afraid there is an expiration date on its contents. If it was timely back in 2019, it is no longer the case in 2025. Many statements are outdated and need to be brought up to the current state of the art. I do not suggest dismissing the authors' attempt to publish an old work; however, I strongly feel that the authors need to rewrite it. In particular, I recommend the following changes to include in the revised version of the manuscript:

  1. Starting with the introduction, the authors refer to an earlier paper of one of the co-authors that introduced the octagon. No further information is provided at this point, and they jump straight to their argument of adopting a "clustering" argument by Komatsu et al. for the current analysis. However, later in the text, in particular in footnote 5 and, mainly, in appendix A, they do indeed offer details on the correlation function analyzed in their work. This presentation, at least, its chronology in the manuscript is not acceptable. It is the authors' prerogative to provide full details in the Appendix, but for the reader's convenience, the definition of the studied observable has to be made in the Introduction with a reference to Appendix A, not for the first time in footnote 5 on page 6 and later in section 3. Footnote 5 needs to be removed since it provides repetitive material. So does footnote 7 in Section 3.

  2. In the paragraph "We thus see ..." on page 3, they mention the scalings of energies/momenta of mirror magnons with (large) 't Hooft coupling. However, this discussion is way too brief and sloppy. While the dominant nature of the mirror plane-wave kinematics for the strong-coupling behaviour of the octagon is known to experts, it might not be the case for the majority of the readership. The authors have to provide a discussion of various domains of the rapidity variable (plane wave, giant magnon, dyonic giant magnon, near flat space), corresponding scalings in 't Hooft coupling as g->infinity, and resulting contributions to the octagon. This will make it comprehensive to a general audience.

  3. The paper mentions multiple times the string dual of the octagon. But it is never made precise in the form of equations. It is no more than a heuristic observation, not grounded in explicit calculations. This has to be formulated explicitly so that the reader realizes that the manuscript does not offer a string dual description for the octagon. In particular, Appendix C does not bring anything of substance to the table: it merely points to an irrelevant (to the current topic) setup when geodesics are not spinning on S^5. If the authors have a better understanding of this topic since the paper was written in 2019, they can offer it there. Otherwise, I insist on removing Appendix C.

  4. On page 8, they say, "Furthermore, if we could compute the one-loop prefactor from the octagon representation, it would provide us with yet another powerful data point to reproduce from the string sigma model." However, I am sure that the authors realize that the calculation of subleading corrections to the octagon from its form in terms of hexagons is extremely hard due to the overlap of various kinematical regions in the rapidity variable. I am not aware of a generalization of the "clustering" analysis beyond the "classical" term. This problem was solved by different means in 2003.01121, to the best of my knowledge, using a complementary representation. If the authors have a clear alternative path to the 1/g expansion, this would be the place to offer their thoughts. Again, they mention in the same paragraph the string side of the story, "it would provide us with yet another powerful data point to reproduce from the string sigma model", but I am not aware of anybody attempting to solve the problem already for the leading term. What is the current status of the problem?

  5. On page 14, there is a statement "To our knowledge, it is the only instance where the ’t Hooft coupling appears explicitly in a physical observable". The authors need to elaborate on the status of this observation since 2019.

  6. The one-loop (from the string perspective) exact nature of eqs. (3.17) is very puzzling! Can the authors explain this? This does not align well with the story of the cusp anomalous dimension. The dual string picture for it is eagerly available, contrary to the octagon. The cusp receives corrections to all orders in 1/g since the GKP background is not supersymmetric. Does the truncation observed in (3.17) imply that for the octagon, there is residual supersymmetry enabling cancellations between various vibrations of the world sheet? How can this be seen, considering the fact that there are no explicit classical solutions to string equations of motion?

  7. The authors posed many questions in their original manuscript, and with Time being the best judge, can they answer any of these after half a decade has lapsed since the original manuscript was written? For instance, "What is this factor of two?" on page 15 was already answered in 2003.01121. There are a few other places that can also be updated, or at least rewritten in light of recent progress or the lack thereof.

  8. Finally, a couple of minor points:

a. Just above (2.9), they say "we only need the strong-coupling expression of the mirror bound-state momentum", but then quote the exact form of the mirror momentum, i.e., Eq. (2.9). The actual approximation is given in (2.11). It is confusing! This needs to be rephrased.

b. The manuscript is full of adjectives like “funny”, “weird”, etc. They are unprofessional and thus not acceptable in a scientific publication. They have to be removed or rephrased!

I cannot recommend this manuscript for publication in its current form. It has to be updated/reworked in light of the above comments to be reconsidered.

Recommendation

Ask for major revision

---

## Round 1 · Referee Report · Anonymous (Referee 2) · 2025-12-23

Strengths

The leading order of the octagon form factor was computed for the first time

Weaknesses

Since the appearance of the preprint in 2019, substantial progress has been made. In particular the whole strong coupling expansion has been found. In order to be published now, the paper should review these developments and complete the list of references.

Report

This paper was contains serious research and was relevant and timely in 2019 when the preprint was posted. Since then, the strong coupling expansion of the octagon has been derived in 2003.01121 and generalized to similar objects in 2207.11475, 2403.13050, 2412.08732. In order to meet the standards for publication in SciPost, the authors should review the progress made since 2019 and cite the relevant papers. Also, the obsolete parts of the manuscript should be removed.

Requested changes

  1. Adjust the Introduction section and give the references for the related papers

  2. Update the Conclusions section

Recommendation

Ask for major revision

---

## Editorial Decision

awaiting_resubmission